# Human gut microbes express functionally distinct endoglycosidases to metabolize the same *N*-glycan substrate

Diego E. Sastre [1] ✉, Nazneen Sultana[1,8], Marcos V. A. S. Navarro[2,9], Maros Huliciak [1], Jonathan Du[1,10], Javier O. Cifuente [3], Maria Flowers [1], Xu Liu[1], Pete Lollar[4], Beatriz Trastoy [5,6], Marcelo E. Guerin [7] & Eric J. Sundberg [1] ✉

Bacteroidales (syn. Bacteroidetes) are prominent members of the human gastrointestinal ecosystem mainly due to their efficient glycan-degrading machinery, organized into gene clusters known as polysaccharide utilization loci (PULs). A single PUL was reported for catabolism of high-mannose (HM) *N*-glycan glyco-polypeptides in the gut symbiont *Bacteroides thetaiotaomicron*, encoding a surface endo-β-N-acetylglucosaminidase (ENGase), BT3987. Here, we discover an ENGase from the GH18 family in *B. thetaiotaomicron*, BT1285, encoded in a distinct PUL with its own repertoire of proteins for catabolism of the same HM *N*-glycan substrate as that of BT3987. We employ X-ray crystal-lography, electron microscopy, mass spectrometry-based activity measure-ments, alanine scanning mutagenesis and a broad range of biophysical methods to comprehensively define the molecular mechanism by which BT1285 recognizes and hydrolyzes HM *N*-glycans, revealing that the stabilities and activities of BT1285 and BT3987 were optimal in markedly different con-ditions. BT1285 exhibits significantly higher affinity and faster hydrolysis of poorly accessible HM *N*-glycans than does BT3987. We also find that two HM-processing endoglycosidases from the human gut-resident *Alistipes finegoldii* display condition-specific functional properties. Altogether, our data suggest that human gut microbes employ evolutionary strategies to express distinct ENGases in order to optimally metabolize the same *N*-glycan substrate in the gastrointestinal tract.

The human gut microbiota is composed of more than 100 trillion microorganisms belonging to around 160 different species, including bacteria, yeast, and viruses. Despite this diversity, microbes from the bacterial phyla Bacteroidales and Bacillota (syn. Firmicutes) represent some 90% of the total gut microbiota[1]. The human gut microbiome has a substantial impact on human health, being essential for the metabolism of complex nutrients, activation of immunity, and protection against colonization by pathogens[2,3]. Symbiotic microorganisms in the human gut provide the enzymatic machinery necessary to degrade a broad range of glycans into their monosaccharide components, which cannot be processed by human intestinal enzymes[4]. Commensal bacteria compete for carbon and energy sources to survive in the densely populated colonic environment[5].

The anaerobic Gram-negative Bacteroidales are abundant mem-bers of the intestinal ecosystem and dominate this niche in part due to their extraordinary ability to efficiently recognize, degrade and import

large, complex and heterogenous carbohydrates through cell surface proteins[6]. The genomes of Bacteroidales contain genes encoding glycan-degrading apparatuses that typically co-localize in discrete clusters known as polysaccharide utilization loci (PULs)[6]. Proteins encoded by these PULs localize to the cell surface and act in concert to bind, degrade, and import diet- and host-derived polysaccharides. PULs are typically defined by their composition as containing SusC-like and SusD-like gene pairs that encode the outer-membrane glycan import machinery, as well as a sensor-regulator, multiple glycosidase hydrolases (GHs) and other carbohydrate-active enzymes that play key roles in the breakdown of specific glycans[6–8]. Often a single PUL encodes the entire apparatus required to process a specific glycan structure, although several PULs containing multiple GHs have been shown to be involved in the degradation of some highly decorated glycans[9–11]. Certain number of PULs have been extensively studied (revised in[12–14]) although still many of the GH structures in complex with their specific substrates have been not determined yet.

The prototypic gut symbiont *Bacteroides thetaiotaomicron* is one of the most prominent members of the human distal gut microbiota and encodes as many as 86 such PULs that collectively encompass approximately 20% of its genome[15–17]. *B. thetaiotaomicron* contains almost 300 different GHs belonging to 60 different GH families, and secretes at least six mannosyl-glycoprotein endo-β-N-acetylglucosaminidases (ENGases, endoglycosidase hereafter) that hydrolyze the chitobiose core of asparagine-linked, or *N*-linked, glycans (EC 3.2.1.96)[18]. This enzyme class comprises GH families 18 (GH18) and 85 (GH85) in the Carbohydrate-Active enZymes (CAZy) Database[19,20], collectively displaying a wide range of glycan specificities processing high mannose (HM), hybrid (Hy), and/or complex-type (CT) *N*-linked glycans. Despite exhibiting the same enzymatic activity, most endoglycosidases within a GH family share only 25-35% sequence identity, suggesting that there are many differences between these enzymes that translate to distinct catalytic mechanisms and divergent substrate specificities.

*N*-glycans are common post-translational modifications of proteins at asparagine residues, particularly on secreted proteins and proteins localized to the surfaces of viruses[21–23]. HM glycans are evolutionarily the oldest class of *N*-glycans and consist of arms of the common pentasaccharide unit, paucimannose, Manα(1-6)-[Man α(1-3)]-Manβ (1-4)-GlcNAc β(1-4)-GlcNAc β1-Asn, with branched arrangements of mannose (Man) monosaccharide units. The HM type *N*-glycans are common in all eukaryotic cells although these have up to nine Man units in mammalian cells, whereas structures can grow to over 200 Man units in yeast mannan[24]. Considering that HM *N*-glycans are intermediates for the eukaryotic glycosylation pathway, they are not commonly found on mature human glycoproteins. However, some HM *N*-glycans are present in relevant human glycoproteins such as immonoglobulins (IgD, IgE, IgM, and IgG)[25], Fc γ Receptors (CD16)[19,26], complement component C3[27], and Galectin3-binding protein, which exhibits high expression levels in the human gastrointestinal tract[28] and elevated induction in metastasis of cholangiocarcinoma[29]. Additionally, there are several glycoproteins in the diet that are rich in HM *N*-glycans, including bovine lactoferricin derived from milk, porcine and calf thyroglobulin, beef pancreas RNAseB, hen egg yolk immunoglobulin (IgY), chicken/hen's egg albumin (ovalbumin), and seed storage proteins from plants, such as 7 S globulins, soybean agglutinin and phaseolins[30]. The highly dynamic nature of HM glycans, as well as their branching complexity and three-dimensional structure, are critical for understanding their roles in immune escape and infectivity of enveloped viruses, such as HIV-1 and SARS-CoV2[21,31,32], and relevant for their abundance on the surface of several types of cancer[33,34] and human embryonic stem cells[35], as well as in the human brain[36].

A single PUL for HM *N*-glycan catabolism in *B. thetaiotaomicron*, which was activated by growth in media containing Man8GlcNAc2 as the sole carbon source, has previously been described[37]. This PUL, known as PUL72, encodes four enzymes and two surface glycan

binding proteins. BT3987 is a surface ENGase, which cleaves the oligosaccharide from its polypeptide. We recently reported the structural basis of glycan specificity by this GH18 endoglycosidase, involved in the processing of mammalian HM and Hy *N*-glycans in the gut[38,39]. The released HM *N*-glycan is held on the cell surface by the mannose-binding protein BT3986, while the SusD homologue BT3984 recognizes its reducing end GlcNAc and orients the glycan into the SusC-like porin BT3983 for transport through the outer membrane. Once in the periplasm, the α−1,2-Man mannosidase BT3990 and α−1,3-Man mannosidase BT3991 release terminal undecorated α−1,6-Man to be hydrolyzed by BT3994, which requires GlcNAc at the reducing end for activity, producing Man-α−1,6-Man-β1,4-GlcNAc. This glycan metabolism pathway results in a trisaccharide that is degraded by enzymes that are not encoded by PUL72. The importance of PUL72 in the metabolism of HM *N*-glycans was considered consistent with the reduced growth of a knock-out deletion mutant lacking the extracellular σ factor regulator of PUL72, BT3993, when cultured in $Man_8GlcNAc_2$[37]. However, growth was restored after some time, suggesting that *B. thetaiotaomicron* can express another ENGase capable of processing HM *N*-glycans, if and when its survival depends on it.

Here, we present the discovery of a HM *N*-glycan-specific ENGase in *B. thetaiotaomicron*, BT1285, which is encoded in PUL16. We show that although BT1285 metabolizes the same HM *N*-glycan substrates as does BT3987, these two endoglycosidases are optimized for distinct conditions. We also define the structural basis by which BT1285 recognizes and hydrolyzes HM *N*-glycans. Furthermore, we found that other bacteria resident in the human gut also encode multiple HM *N*-glycan-specific ENGases in PULs analogous to *B. thetaiotaomicron* PUL72 and PUL16 that respectively encode BT3987 and BT1285. Together, these data suggest that individual human gut microbes express multiple functionally distinct endoglycosidases capable of metabolizing the same *N*-glycan substrate in order to survive across a wide range of nutrient availability and environmental conditions.

## Results

### Bacteroides thetaiotaomicron encodes two high-mannose glycan-specific endoglycosidases

To define sequence-structure-function relationships within the GH18 family of ENGases and to predict substrate specificity, we performed sequence similarity network (SSN) analysis[40] of all GH18 family members contained in the CAZy database (more than 53 thousand sequences)[41]. We found that the GH18 family of ENGases (EC 3.2.1.96) from the prevalent human gut bacterium *B. thetaiotaomicron* VPI-5482 strain could be classified into four subgroups or clusters, containing only phylum Bacteroidales-derived enzymes with an alignment score threshold of 50 (Fig. 1a, Supplementary Fig. 1a). Proteins in two of these clusters (clusters 1 (C1) and 4 (C4); composed of 77 and 4 open reading frames (ORFs), respectively) contain a domain of unknown function called DUF4849, and include the *B. thetaiotaomicron* GH18 putative ENGases BT1038, BT1044, BT1048, BT3753, BT4406 in C1, as well as BT4709 in C4. The other two clusters (C2 and C3; composed of 176 and 18 ORFs, respectively) lack this domain, and include BT3987 in C2 and BT1285 in C3. All of these proteins are predicted to exhibit the typical $(β/α)_8$- or TIM-barrel fold of GH18 catalytic domains, as supported by reported crystal structures of homologues[11,38] or Alphafold2 structure predictions[42] (Supplementary Fig. 1b). Members of C1 and C4, which contain DUF4849 (PF16141), were previously shown to be involved in complex-type (CT) *N*-glycan catabolism[11]; those in C2, including EndoF1[43], EndoH and BT3987[38], are able to specifically process HM *N*-glycans and members of C3, including BT1285, show higher sequence similarity to cluster C2 than the other clusters. BT1285 and BT3987 are encoded in two different PULs[17]: PUL16 comprises genes encoding for the BT1278 to BT1285 proteins; whereas PUL72 contains genes encoding for the BT3983 to BT3988 proteins (Fig. 1b). Expression of the gene products in these PULs are each regulated by independent

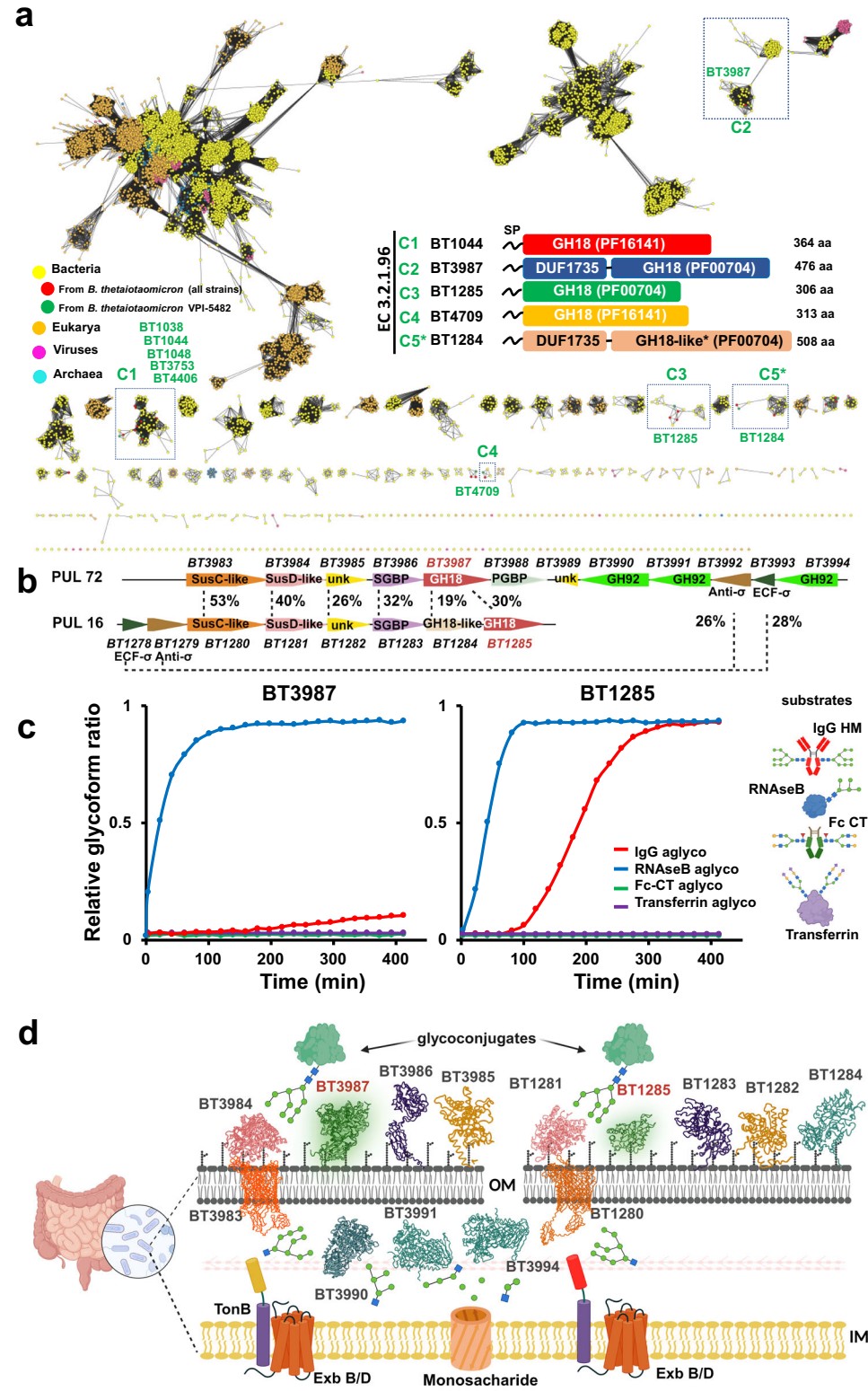

**Fig. 1 | *Bacteroides thetaiotaomicron* encodes two high-mannose glycan-specific endoglycosidases. a** Sequence similarity network of GH18 family from CAZY database at at a alignment score threshold of 50, representing 10,984 protein nodes. **b** Schematic representation of PUL16 and PUL72 of *B. thetaiotaomicron*. ECF σ, extracytoplasmic σ factor; SGBP, surface glycan binding protein; unk, protein with unknown function. Percentage of protein sequence identity between the syntenic genes is indicated. **c** Competitive kinetic analysis of BT3987 (50 nM) and BT1285 (50 nM) in the presence of four different glycoprotein substrates (2 µM each). Assays were run in technical triplicates ($n = 3$). Curves represent the complete deglycosylated substrate proportion. **d** Schematic representation of HM specific PULs in *B. thetaiotaomicron* indicating predicted subcellular localization of protein components of PUL 16 and 72. Cartoon representation of protein structures were obtained from AF models, with the exception of BT3984 (pdb: 8UWV), BT3984 (pdb: 6TCV), BT3990 (pdb: 2WVX) and BT1281 (pdb: 4MRU). Sugar symbols: GlcNAc, Man, Fuc, Gal, Neu5A. Source data are provided as a Source Data file.

extracellular sigma factors and anti-sigma factors, and contain SusC- and SusD-like proteins (Fig. 1b).

To further investigate the activities and substrate specificities of the GH18 family members found in PUL16 and PUL72, we performed Competitive-Specificity of Enzyme Activity with Kinetics (C-SEAK) assays[44] using a mixture of four different substrates: (1) an Fc region of IgG1 containing CT N-glycans attached to Asn297 of both protomers; (2) RNAseB, an endoribonuclease that presents a single N-linked glycosylation site at N34 bearing HM N-glycans (mainly the HM glycans $Man_5$ and $Man_6$); (3) Transferrin, a protein that contains sialylated bi- and tri- antennary CT N-glycans; and (4) a monoclonal IgG1 antibody (Rituximab)-HM, bearing $Man_8$ and $Man_9$ at Asn297. In the kinetic analyses shown in Fig. 1c, we found that both BT1285 and BT3987 only process the HM N-glycan attached to RNAseB and IgG substrates and not the CT N-glycans on Transferrin and IgG, confirming that they are HM-specific GH18 ENGases. Additionally, we observed that BT1285 was comparable to BT3987 for hydrolysis of RNAseB, but it was nearly 20-fold faster than BT3987 for hydrolysis of IgG-HM (Fig. 1c). Thus, both BT1285 and BT3987 are likely localized at the outer leaflet of the outer membrane to cooperatively and specifically metabolize and import HM N-glycans into the B. thetaiotaomicron cells (Fig. 1d).

We found that PUL16 also encodes another putative GH18 endoglycosidase variant, BT1284. However, this protein contains non-conservative substitutions of both acidic residues in the catalytic site, thereby lacking the canonical catalytic DGxxxDxE motif of bona fide ENGases (Supplementary Fig. 2a). We found ENGase-like proteins homologous to BT1284 throughout several Bacteroidales species (85 sequences in at least 20 different species) with similar substitutions of catalytic (Supplementary Fig. 2a) and substrate binding site residues. By enzymatic activity measurements using complex type glycans (from both animal and plant sources) and HM N-glycans as substrates, we confirmed that BT1284 and one ortholog from Bacteroides faecium (Uniprot ID: A0A6H0KVH8), which we named as B. faecium GH18-like, were indeed catalytically-impaired (Supplementary Fig. 2b–d). This impaired ENGase activity of BT1284 and other GH18-like proteins throughout Bacteroidales species is similar to that of GH18-like proteins with impaired chitinase activity, some of which are known to function as lectins and play a key role in the physiology of plants and humans[45–47]. We also assessed the HM N-glycan binding capacity of BT1284 by SPR analyses. However, we observed no measurable binding of BT1284 for HM N-glycan substrates (neither RNAseB nor IgG HM) (Supplementary Fig. 2e). To better understand their lack of catalytic activity and glycan binding ability, we determined the X-ray crystal structure of B. faecium GH18-like to a maximum resolution of 2.67 Å (Supplementary Fig. 3 and Supplementary Table 4). By using the DALI server[48] and FoldSeek[49], we found that the closest structural homologs of B. faecium GH18-like are BT3987 (root-mean-square deviation (RMSD) of 2.9 Å in 435 residues, 19% ID)), EndoH (23%ID) and EndoF1 (18% ID), all HM-specific ENGases. The crystal structure of B. faecium GH18-like revealed the presence of a TIM barrel fold typical of ENGases from GH18 family fused to a β-sandwich domain in the N-terminal region (DUF1735), similar to BT3987 (Supplementary Fig. 3), as well as a β-hairpin motif common to GH18 enzymes with specificity towards HM N-glycans (Supplementary Fig. 1b). Structural surface analysis in the putative glycan binding pocket of this ENGase-like protein, revealed that HM N-glycans do not fit as well as in a bona fide HM endoglycosidases (Supplementary Fig. 3e). Together, these data indicate that ENGase(GH18)-like proteins were unable to hydrolyze or bind N-glycans in the assayed conditions, but may they still play a role in the metabolism of N-glycans in HM-PULs from Bacteroidales by some as yet determined mechanism.

## Structural basis for the HM N-glycan substrate specificity of BT1285

To determine the mechanism by which the discovered HM-processing GH18 ENGase BT1285 recognizes and hydrolyzes N-glycans in comparison to its previously described counterpart BT3987[38], we first determined the X-ray crystal structure of BT1285 alone at 1.08 Å resolution. BT1285 displays the typical $(\beta/\alpha)_8$-barrel catalytic domain found in other GH18 family members (Supplementary Fig. 4 and Supplementary Table 3). By using the DALI server[48] and FoldSeek[49] we found the closest structural homologs to be EndoH (PDB code 1c8y[50]; Z-score, 37.2; r.m.s.d of 1.3 Å for 253 aligned residues; 36% identity), and BT3987 (Z-score, 31.7; r.m.s.d. of 2.4 Å for 263 residues; 29% identity), in agreement with its ability to efficiently deglycosylase HM N-glycans. To better understand the molecular mechanism by which BT1285 specifically hydrolyzes N-glycans, we solved the crystal structure of a catalytically inactive variant of BT1285 ($BT1285_{D161A-E163A}$; hereafter BT1285i) in complex with the $Man_9GlcNAc_2$ substrate at 1.9 Å resolution (Fig. 2a–d and Supplementary Fig. 5). We found one molecule of the $Man_9GlcNAc_2$ substrate unambiguously positioned in the active site of the enzyme flanked by the following connecting regions: β1-α1 (loop 1; residues 52–57), β2-α2 (beta hairpin; residues 78-94), β4-α3 (loop 2; residues 123–135), β5-α4 (loop 3; residues 161-178), β6-β7 (loop 4; residues 199-204), β7- β8 (loop 5; residues 218-240), β8-α5 (loop 6; residues 243-250), β9-α6 (loop 7; residues 272–282) (Fig. 2a, c, d).

We observed no substantial conformational changes in the protein backbone for the unliganded and glycan-bound forms of BT1285 (r.m.s.d. of 0.4 Å for 271 residues). The main structural differences between BT1285 and BT3987 involve the length of the β-strands of the TIM barrel (particularly the β2 strand) and the presence of longer flexible loops in BT1285, with higher B-factor values around active site in BT1285 relative to those in BT3987, which decreased in the presence of the HM substrate (Fig. 2b and Supplementary Fig. 6a). The overall architecture around the active site and substrate binding pocket exhibit an RMSD of 2.4 Å between these HM-processing ENGases (Supplementary Fig. 6b). For example, Man (−5) forms hydrogen bonds with residues E166 and R169 in BT1285 while it is exposed to solvent in BT3987, and Man (−9) is exposed to the solvent in BT1285, but it is in close contact with residue N403 in BT3987 (Fig. 2d and Supplementary Fig. 6c), These observations are consistent with the degrees of flexibility displayed by Mannoses within the crystal structures of both ENGases bound to HM-N-glycan, as evidenced by the B-factor values (Supplementary Fig. 6d).

To further investigate how BT1285 specifically recognizes HM substrates at the molecular level and how it compares to other HM-processing ENGases such as BT3987, we performed an alanine scanning mutagenesis analysis of residues that decorate the β-barrel core of the enzyme and contact the $Man_9GlcNAc_2$ N-linked glycan substrate. We determined the ability of each single-site alanine mutant to process HM N-glycans on the HM-IgG substrate. Specifically, we mutated residues in loop 1 (E52A, N54A, D55A), the β-hairpin loop (N81A and N94A), loop 2 (H126A), loop 3 (E166A and R169A) and loop 5 (F222, which interacts with the GlnNAc core). As shown in Fig. 2e, E52A from loop 1, H126A from loop 2, and E166 from loop3 drastically reduced the hydrolytic activity against IgG of the enzyme (at 37 °C, pH 7.4). Mutations in residues N81, N94 and R169 also reduced enzymatic activity but to a lesser extent. Simultaneous mutation of two non-catalytic residues, Glu52 and His126, both of which engage the N-glycan core (Fig. 2f and Supplementary Fig. 7), completely abolished BT1285 hydrolytic activity, indicating that together these two residues are essential for HM substrate recognition. Collectively, our mutational analysis of the BT1285 residues that contact the HM glycan indicated that interactions with the Man (−2), Man (−3), Man (−5) and Man (−6) saccharides in the core and antennae A and B, were critical for glycan recognition, while those with antenna C (α1,3) saccharides were dispensable for substrate recognition (Fig. 2f, Supplementary Figs. 6 and 7). Moreover, this is in agreement with the apparent capacity of BT1285, akin to BT3987, to also bind to triantennary hybrid-type N-glycans because the solvent exposure of GlcNAc and Gal sugars

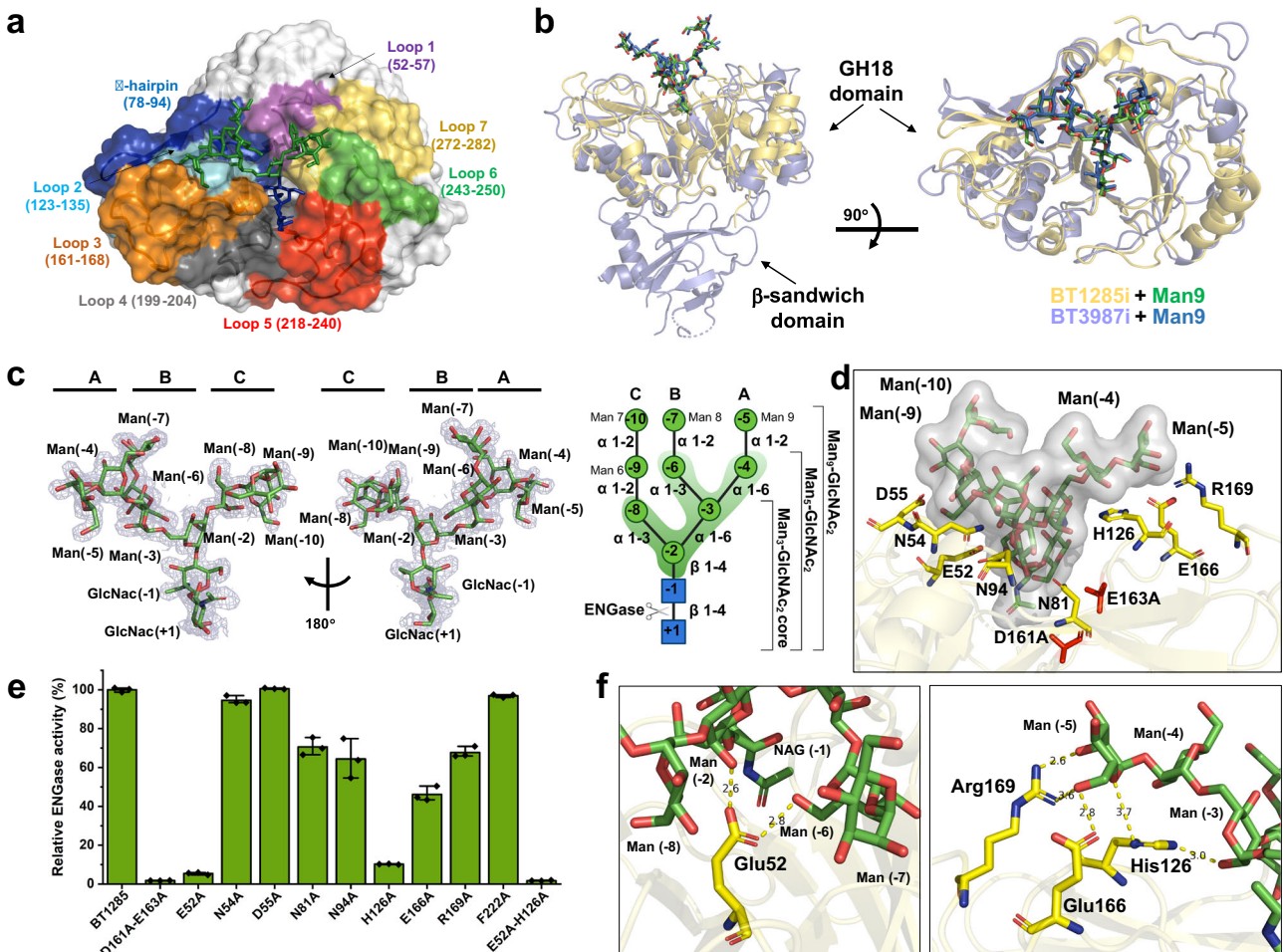

**Fig. 2 | Structural basis for the HM *N*-glycan substrate specificity of BT1285.**
**a** Surface representation of the crystal structure of BT1285_{D161A-E163A} (BT1285i) in complex with Man₉GlcNAc₂ (pdb code: 8U48). The different GH loops around the HM N-glycan are distinctly colored. **b** Two views of cartoon representation of superimposed crystal structure of BT1285i (yellow) and BT3987_{D312A-E314L} (BT3987i) (purple) (pdb code:6TCV[38], both in complex with Man₉GlcNAc₂. **c** Two views of the electron density map of Man9GlcNac2 substrate from BT1285_{D161A-E163A} -Man₉GlcNAc₂ crystal structure, shown at 1.0 σ r.m.s deviation. In left part, schematic representation of Man9GlcNac2 substrate. Sugar symbols: ▪

GlcNAc, 🟢 Man. β-1-4 bond cleaved by ENGases is indicated with scissors. **d** Close inspection of substrate binding site of BT1285 in complex with Man₉GlcNAc₂. Residues used in the alanine scanning assays are represented as sticks. **e** Hydrolytic activity of BT1285 and mutants against HM-IgG1, as determined by LC-MS analysis, normalized to BT1285 wt. Data are presented as mean values±SD. Assays were run in technical triplicates (*n* = 3). **f** Close inspection of key residues involved in the Mannoses recognition and binding in BT1285. Source data are provided as a Source Data file.

attached to antenna C (Supplementary Fig. 7). Furthermore, we verified that HM-specific ENGases, BT1285 and BT3987 are unable to accommodate neither complex-type *N*-glycan structures with α1,6-fucose attached to the first core GlcNAc (present in mammalian complex-type *N*-glycans) nor α1,3-fucose attached to the core GlcNAc that are decorations in plant *N*-glycans (Supplementary Fig. 2c). While all these results are similar to what has been described for BT3987[38], the interaction with Man (−2) and Man (−6) by BT3987 residue Glu200 (analogous to Glu52 in BT1285) did not affect its catalytic activity as much as we observed in the mutant E52A in BT1285.

## Stabilities and activities of BT1285 and BT3987 are differentially affected by pH and temperature

To gain further insight into why *B. thetaiotaomicron* VPI-5482 encodes two GH18 ENGases with identical *N*-glycan specificities, we assessed the functional properties of BT3987 and BT1285 over a range of both temperature and pH. First, we analyzed their thermostabilities at different pH levels using differential scanning fluorimetry (DSF; Fig. 3a). We found that BT3987 exhibited peak stability at pH 6.0 and was especially unstable at low pH, while the stability of BT1285 was largely unaffected by pH. Furthermore, BT1285 was more stable than BT3987 at all pH levels.

Next, we determined the impact of pH on the catalytic activities of these endoglycosidases. Using intact glycoprotein mass spectrometry (Fig. 3b), we found that hydrolysis of HM *N*-glycans on RNAseB by BT3987 was optimal at pH levels between 6.0 and 8.0, with substantially reduced activity at both lower and higher pH, similar to the effect of pH on its thermostability. Conversely, BT1285 was highy active across a broad range of pH levels and was increasingly active with acidic conditions, different of most previously characterized ENGases which exhibited an optimal pH around 4.5–7.5 (e.g., EndoH, EndoF1, F2, F3 EndoS and EndoS2)[25,51].

We identified two aromatic residues surrounding the catalytic residues that are interchanged between the two *B. thetaiotaomicron* HM-ENGases−Tyr315 in BT3987 is in the analogous position as Trp164 in BT1285, while Trp355 in BT3987 is in the analogous position as Tyr201 in BT1285 (Fig. 3c and Supplementary Fig. 8). By exchanging these residues in BT1285, via a BT1285_{W164Y-Y201W} mutant, we observed an activity profile more similar to that of BT3987 than BT1285 wild type (Fig. 3d), although these residues do not completely explain the differences observed in the catalysis performed by BT1285 and BT3987. We also found that a His126 in loop 3 (but not Arg169, Supplementary Fig. 8) affects the optimal pH of BT1285 (from pH 2 to pH 4).

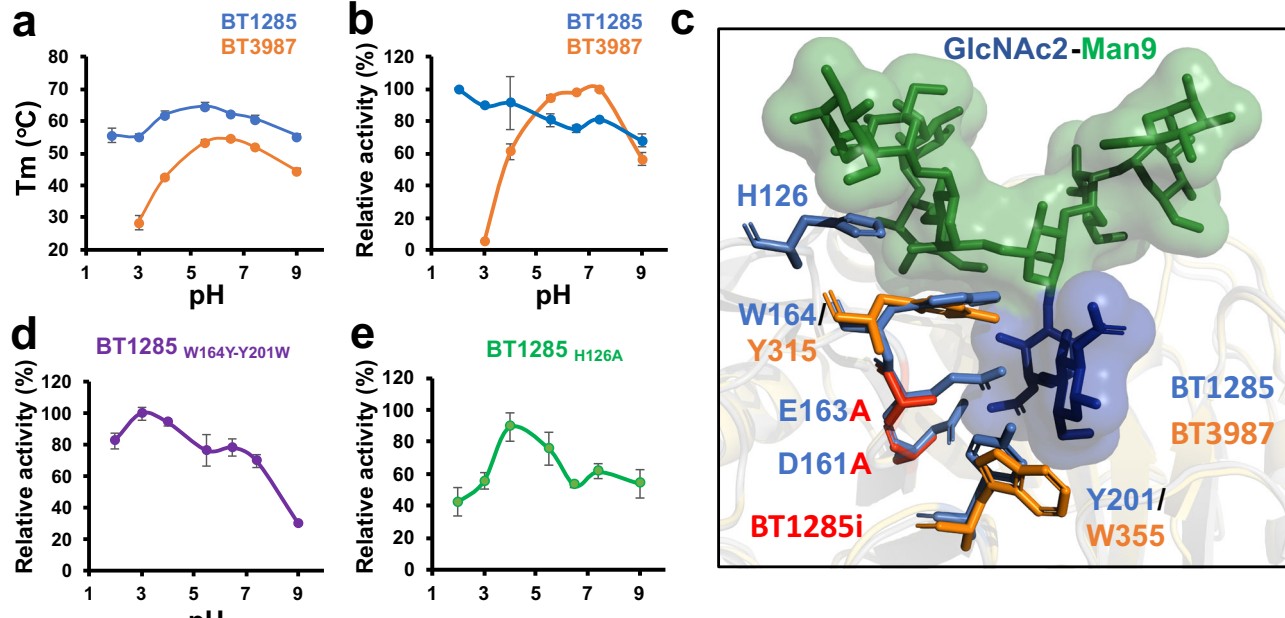

**Fig. 3 | BT1285 and BT3987 enzymes are differentially affected by temperature and pH changes. a** Thermal stability of BT1285 (blue line) and BT3987 (orange line). Assays were run in technical triplicates ($n = 3$). Data are presented as mean values ± SD. **b** Relative activity mesearuments of BT1285 and BT3987 wt enzymes using RNAseB as a substrate at different pH in the range of 2 to 9 at 23 °C. ($n = 3$ technical triplicates) Data are presented as mean values ± SD. **c** Close inspection of cartoon representation of crystal structure of BT1285 in complex with Man$_9$GlcNAc$_2$ (pdb code: 8U48) showing the differential aromatic residues with BT3987 and possible residues involved in pH dependence. Active site is composed by D161 and E163 residues. **d**, **e** Relative activity of BT1285 mutants measurements using RNAseB as a substrate at different pH in the range of 2 to 9. Endpoint assays were performed by techinical triplicates ($n = 3$). Data are presented as mean values ± SD. Source data are provided as a Source Data file.

Specifically, His126, which engages the *N*-glycan substrate (Fig. 3e), likely changes the optimal pH of BT1285 by influencing the pKa values of the key acidic active site residues (Asp161/Glu163), as has been previously reported in acidic chitinases from the GH18 family[52,53] and in the arginine deiminase from *P. aeruginosa*[54].

**Conformational accessibility of *N*-glycans affects their recognition and hydrolysis by BT1285 and BT3987**

The major difference between the two HM glycoprotein substrates used in these studies, IgG-HM and RNAseB, is the accessibility of the HM *N*-glycan. In RNAseB, the HM *N*-glycan is highly exposed, as it is linked to an asparagine residue in a loop. Conversely, the HM *N*-glycans on IgG are linked to the Asn297 residues on each of the two IgG heavy chain protomers; these glycans face inward towards the center of the IgG dimer and require protein conformational changes to become accessible for deglycosylation by endoglycosidases[55,56]. As shown in Fig. 1c, both BT1285 and BT3987 hydrolyzed HM *N*-glycans on RNAseB rapidly, but BT1285 hydrolyzed HM *N*-glycans on IgG-HM much faster than BT3987. Thus, we assessed further how the conformational accessibility of the HM *N*-glycan substrate affects BT1285 and BT3987 glycan recognition and hydrolytic activity.

To determine the binding affinities of BT1285i and BT3987i to *N*-glycans, we performed SPR analyses using (i) a Protein A sensor chip on which we captured IgG1 (Man$_9$, Man$_5$ or CT) or (ii) a Carboxy-methylated Biacore CM5 Sensor Chip onto which RNAseB in its glycosylated form was immobilized on flow cell 2, and aglycosylated RNAseB version was immobilized on the control flow cell 1. We analyzed the association ($k_{on}$) and dissociation ($k_{off}$) rate constants and calculated the dissociation constant $K_D$ for each interaction whenever possible. Our data clearly confirms the specificity of both ENGases, BT1285 and BT3987, for HM substrates, as none of them were able to bind to CT *N*-glycans on IgG1 (Fig. 4). Indeed, we observed no measurable binding of BT3987i for any IgG substrate regardless of whether it displayed HM or

CT *N*-glycans. While both BT1285i and BT3987i inactive ENGases bind to the Man$_5$ oligomannose on RNAseB, BT1285i exhibited a 1000-fold higher binding affinity than BT3987i ($K_D$s of 3.5 nM versus 4.3 µM). Therefore, glycan accessibility clearly plays a role in these binding events, as the affinity of BT1285i for the exposed glycan on RNAseB was 12-fold higher than for the constrained glycans on IgG ($K_D$s of 3.5 nM versus 42 nM). We also determined how the H126A mutation affects the binding affinity of BT1285i to HM-glycans (Supplementary Fig. 9). As expected, BT1285i$_{H126A}$ exhibited a 1000-fold lower affinity than did BT1285i to all HM *N*-glycans tested (Supplementary Fig. 9), regardless of the glycoprotein on which they were displayed – RNAseB, IgG-Man$_9$ and IgG-Man$_5$ – confirming the importance of His126 for binding to the oligomannosidic substrate (Fig. 2e and 2f).

To further define why BT1285 has such a catalytic advantage over BT3987 in hydrolyzing HM *N*-glycans from IgG substrates, we assessed the kinetics of hydrolysis of both glycans on the IgG homodimer. With one glycan on each of the Asn297 residues of the two IgG protomers, the possible glycosylation species that exist include: bi-glycosylated (Asn297-linked glycans on both IgG protomers; the initial substrate), mono-glycosylated (an Asn297-linked glycan on only one IgG protomer; an intermediate in the full deglycosyltion reaction and also a substrate for further deglycosylation), and deglycosylated (no Asn297-linked glycan on either protomer; the final product) (Supplementary Fig. 10). As shown in Fig. 5a and Supplementary Fig. 10, the reaction rates of IgG-HM deglycosylation by BT3987 and BT1285 appear especially distinct with all three species−substrate, intermediate and product−observed over time. For BT3987, we observed an accumulation of mono-glycosylated IgG-HM even after 400 minutes, while for BT1285 the mono-glycosylated IgG-HM species appeared rapidly, peaked at 75 min, and was completely hydrolyzed into fully deglycosylated IgG-HM by 200 min. The symmetry of the mono-glycosylated intermediate peak in these assays suggests that BT1285 has no preference for either the di-glycosylated

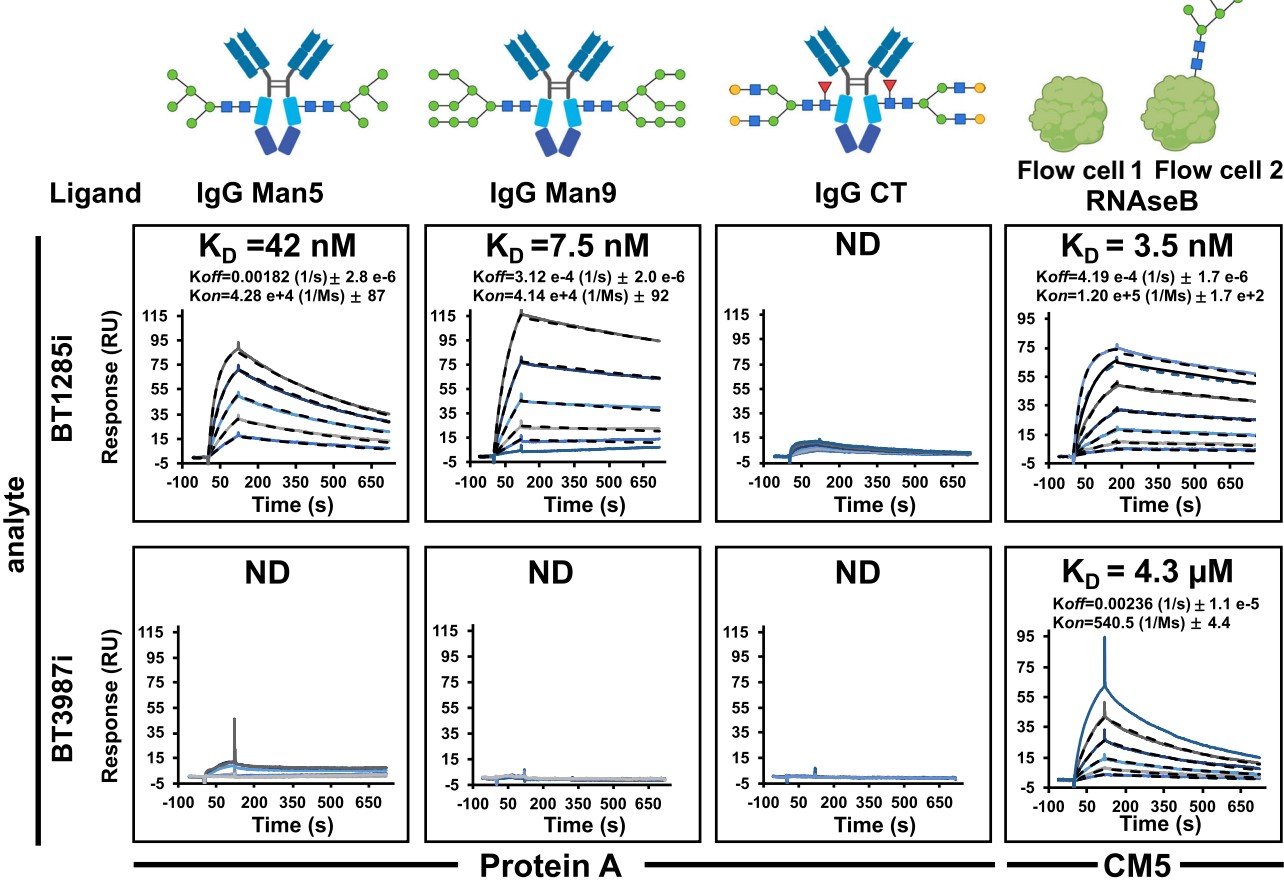

**Fig. 4 | Binding affinities of BT1285i and BT3987i to different *N*-glycan substrates by SPR.** In the upper panel are represented the binding kinetics of 1285i (analyte) vs different *N*-glycan substrates immobilized to a protein A (Man₅, Man₉, CT) and CM5 sensor chip (RNaseB) using amine coupling to a surface density of 100–150 response units (RU). In bottom panel shows binding analysis of BT3987i (analyte). Concentration range of BT1285i analyte plotted (serial dilution 1:2): 37.5 nM to 600 nM for IgG-Man₅/Man₅; 9 nM to 300 nM for IgG Man₉/Man₉; 156 nM to 5 µM for IgG CT, and 3.75 to 240 nM for RNaseB. For BT3987i analyte concentration range plotted (serial dilution 1:2): 625 nM to 20 µM for IgG Man₅/Man₅; 156 nM to 5 µM for Man₉/Man₉ and CT; 1 µM to 32 µM for RNaseB. Black dashes line represent the fitting on kinetic models. ND=not determined K_D. Assays were run in independent duplicates (*n* = 2). Sugar symbols: ■ GlcNAc, ● Man, ▲ Fuc, ● Gal. Source data are provided as a Source Data file.

or mono-glycosylated IgG-HM species. To further verify that the mono-glycosylated IgG-HM intermediate is not a specific target of BT1285 hydrolysis, we performed SPR assays using protein A sensor chips in which di-glycosylated, mono-glycosylated (on each of both protomers), or deglycosylated IgG-HM species were immobilized and BT1285i was introduced as the analyte. As shown in Fig. 5b, we observed similar affinity and kinetics for the interaction of BT1285i with the di-glycosylated and mono-glycosylated IgG-HM. These experimental data also suggest that BT1285 does not sense any particular conformational or dynamic change in IgG that could result from the hydrolysis of one glycan and supports the mechanism by which BT1285 exclusively recognizes the HM *N*-glycan on IgG.

To determine whether the β-sandwich domain of BT3987, which is absent in BT1285, was responsible for its relatively reduced hydrolytic activity, we compared the kinetics of IgG-HM hydrolysis of full-length BT3987 with its GH18 enzymatic domain alone (Supplementary Fig. 11). We found that these enzymes exhibited similar kinetics, indicating that the BT3987 β-sandwich domain was not responsible for the slower hydrolytic rates of BT3987, although the lack of this domain drastically reduced the thermal stability of this enzyme (Tm 52 °C vs 41 °C at pH 7.4; Fig. 3a and Supplementary Fig. 11b). Using the same antibody substrate but changing the HM *N*-glycan from Man₉ to Man₅, we obtained similar results, indicating that the size of the glycan was also not responsible for the difference in hydrolytic rates (Supplementary Fig. 11).

## Protein-protein interactions in the BT1285i/Fc-HM complex are dispensable for catalytic activity

Taking into account (i) the high affinity observed between BT1285i and IgG1-HM and (ii) the substantially faster kinetics of HM *N*-glycan hydrolysis by BT1285 on this substrate, relative to BT3987, we sought a structural basis for these activities. By using size exclusion chromatography (SEC) we determined that BT1285i forms a stable complex with either Fc-IgG1-HM (Fig. 6a) and IgG1-HM, but not with IgG-CT as expected (Supplementary Figs. 12 and 13). SDS-PAGE analysis of the SEC fractions indicated the formation of a 2:1 BT1285i:Fc-HM highly stable complex (Fig. 6a). We validated the 2:1 stoichiometry of such a complex by both analytical ultracentrifugation (AUC) (Fig. 6b) and SEC-multiple angle light scattering (MALS) (Supplementary Fig. 12). The SEC-MALS and AUC estimates a MW of approximately 104 kDa, slightly lower than the predicted 116 kDa from the amino acid sequences. These discrepancies may be due to the dynamic equilibrium between free BT1285 and Fc so the peaks in both SEC-MALS and AUC contain Fc dimer bound to a single BT1285. Overall, the results of the AUC and SEC-MALS experiments are consistent with high affinity binding of BT1285 to the Fc dimers at ratios of 2 BT1285 molecules per Fc dimer. We next performed small-angle X-ray scattering (SAXS) with in-line SEC and reconstructed an ab initio low-resolution envelope of the BT1285i:Fc-HM complex (ca. 125 kDa, Supplementary Fig. 14, Supplementary Table 5), in which we

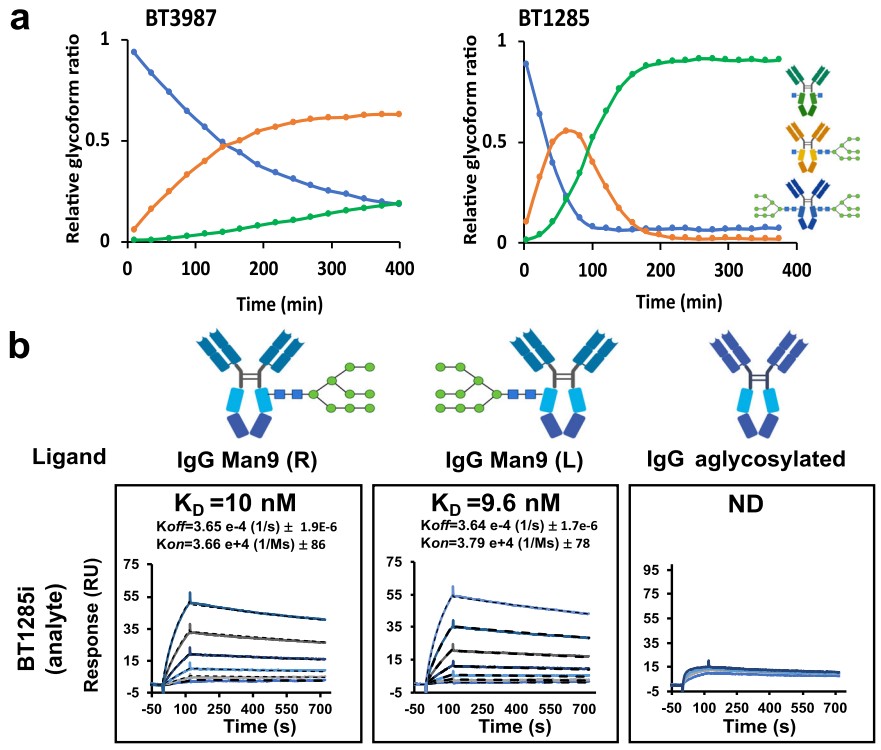

**Fig. 5 | Kinetics of hydrolysis and binding of BT1285 to both glycans on the IgG homodimer. a** Kinetic analysis of BT3987 and BT1285 vs IgG HM. Asssay is representative of triplicates at 37 °C in HBS buffer. **b** SPR assays of BT1285i (analyte) with different diglycosylated, monoglycosylated and aglycosylated (N297Q) IgG variants. Concentration range of analyte was 4.7 nM to 300 nM for monoglycosylated IgGs, and 156 nM to 5 μM for IgG deglycosylated. Assays are representative of two independent experiments. Sugar symbols: ■ GlcNAc, ● Man. Source data are provided as a Source Data file.

fit two copies of our crystal structure of BT1285 and one copy of an Fc (Fig. 6c). In addition, we obtained a native stable complex between BT1285i and Fc HM at low protein concentration (0.05 mg mL⁻¹) that we visualized by negative-stain transmission electron microscopy (NS-TEM), which did not require chemical crosslinking to be stabilized as previously reported for the EndoS-Fc complex[57], due to the high affinity of BT1285 for HM glycans on Fc. By NS-TEM analysis of the BT1285i:Fc-HM complex we generated a three-lobed ~19 Å structure, in which one molecule of BT1285 is positioned to attack the Asn297-linked glycan on each Fc protomer and making contact just with the HM N-glycan on N297, which also supported the 2:1 BT1285i:Fc-HM stoichiometry (Fig. 6d, Supplementary Figs. 15 and 16). Despite the limited resolution of our BT1285i:Fc-HM complex structure, we observed a likely protein-protein interface in this complex (Fig. 5e). To test whether these protein-protein interactions contribute to the enzymatic activity of BT1285, we mutated key residues in the putative interface from both BT1285 and Fc-HM, individually or in combination, and measured their enzymatic activities (Fig. 5f).

Compared to wild type BT1285, we found no mutant with a significantly different hydrolytic rate, suggesting that this protein-protein interaction does not contribute to the high catalytic rate of BT1285 and that only recognition of the HM N-glycan itself is driving catalysis, distinct from what we previously observed in the IgG-specific ENGase EndoS[57] in which the protein-protein contact between IgG-EndoS was essential for catalysis.

### Numerous gut bacteria species encode multiple ENGases that hydrolyze the same HM glycan substrates

Having shown that a single human gut microbe, *B. thetaiotaomicron* VPI-5482, can produce two GH18 ENGases with the same HM N-glycan substrate specificity but with distinct functional properties, we asked whether such a phenomenon may be more widespread amongst

bacteria that colonize humans. Accordingly, we bioinformatically explored PULs across the entire Bacteroidales phylum. By modular alignment analysis, we found several species containing at least two different HM-processing ENGases located in distinct PULs, including but not limited to *B. faecium*, *B. ovatus*, *B. xylanisolvens*, *Alistipes finegoldii* and *A. onderdonkii* (Supplementary Fig. 17). This suggests that encoding multiple endoglycosidases with the same N-glycan specificity may be common. We selected two ENGases from *A. finegoldii*, a relatively recently assigned sub-branch genus of the Bacteroidales phylum associated with dysbiosis and several intestinal diseases in humans[58], for further analysis. Alfi_0882, which we found in C2 of our SSN analysis (Fig. 1a and Supplementary Data 1) shares a similar overall architecture with BT3987 (48% sequence identity with BT3987 versus 24% sequence identity with BT1285), whereas Alfi_0894, which we found in C3 (Fig. 1a and Supplementary Data 1), lacks the β-sandwich domain on its N-terminus like BT1285 (38% sequence identity with BT1285 versus 25% sequence identity with BT3987). Additionally, the genes encoding Alfi_0882 and Alfi_0894 are located in PULs that are analogous to the *B. thetaiotaomicron* PULs encoding BT3987 and BT1285, respectively (Fig. 7a).

We expressed and purified recombinant versions of Alfi_0882 and Alfi_0894 and conducted C-SEAK analysis with these enzymes and our standard panel of four glycoprotein substrates: Fc-CT, RNAseB, Transferrin and IgG-HM (Fig. 7b), as well as SEAK analysis with IgG-HM as the sole substrate (Fig. 7c). We confirmed that both Alfi_0882 and Alfi_0894 were HM-processing ENGases. Alfi_0882 exhibited fast kinetics with both HM glycoprotein substrates, RNAseB and IgG-HM, while Alfi_0894 exhibited fast kinetics only with RNAseB and slow hydrolysis of IgG-HM. In this regard, Alfi_0882 is similar to BT1285, while Alfi_0894 is akin to BT3987. We also measured thermal stability (Fig. 7d) and hydrolytic activity (Fig. 7e) across a wide pH range and found that the functional properties of Alfi_0882 were generally similar to those of BT3987, while for Alfi_0894 they were most alike to those of

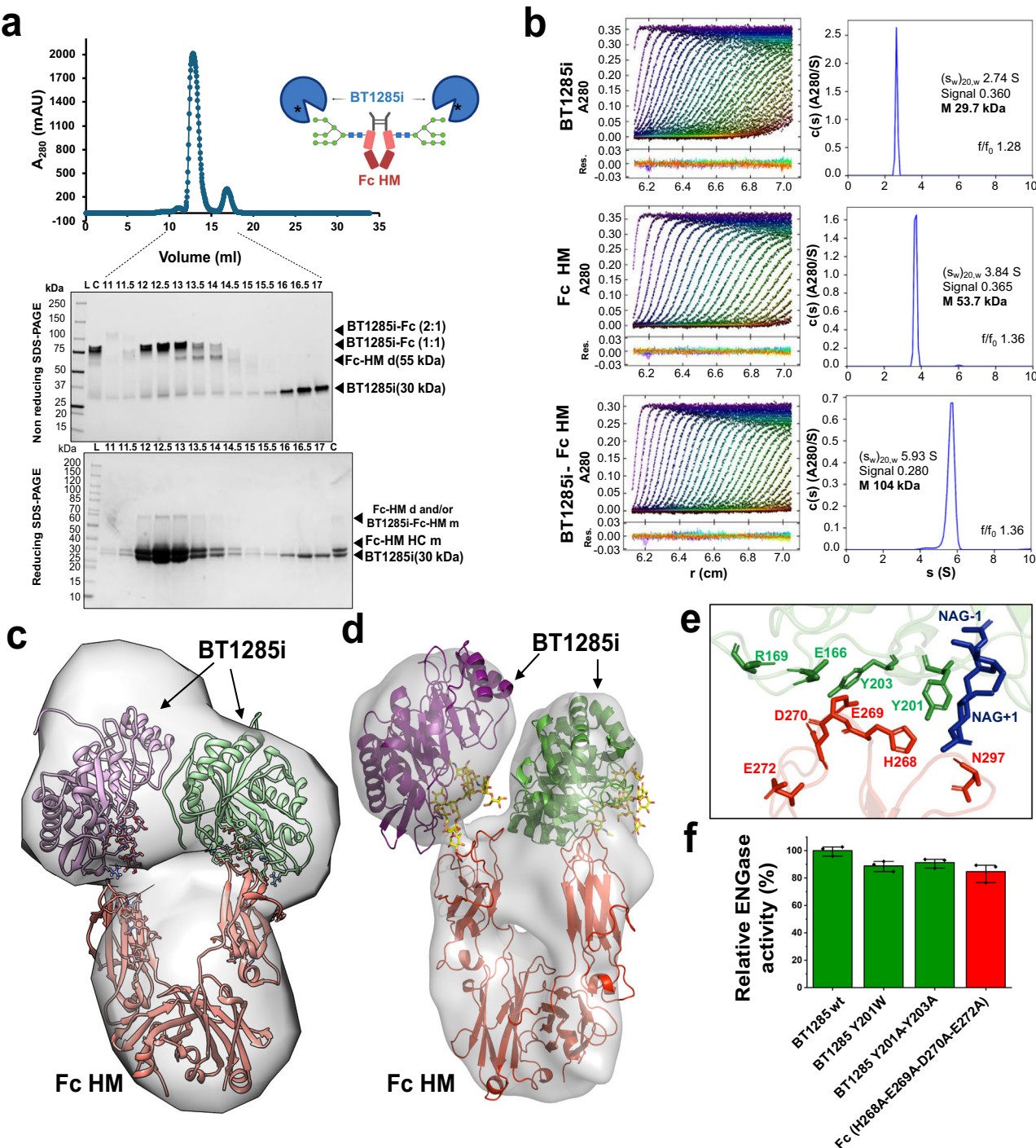

**Fig. 6 | Analysis of BT1285i-Fc HM complex formation. a** SEC analysis of BT1285i-Fc complex run on Superdex S200. Reducing and non-reducing free-stain SDS-PAGE (4–20%) analysis of eluted fraction from gel filtration. Indicated with letter L (ladder), C (complex, fraction injected into the SEC). Experiment were repeated independently three times with similar results. **b** AUC analysis of BT1285i, Fc HM and BT1285-Fc complex (fraction correspond to fraction eluted in volume 12.5 ml in SEC in Fig. 6a. Samples in HBS Buffer were centrifuged at 45,400 g. Curves represent fits to the continuous c(s) distribution model in SEDFIT. Res.=residuals. **c**, **d** 3D reconstruction of SAXS **c** and NS-TEM **d** data, respectively, revealed 2:1 complex low-resolution structures. We fiited high-resolution structure of Fc (pdb code:5JII) and BT1285i-HM (pdb code:8URA) in SAXS and NS-TEM maps. **e** Close inspection of BT1285i-Man$_9$-Fc interface region showing main resdiudes of both proteins as sticks. **f** Relative ENGase activity measurements of selected mutants in the interface region of Fc (red) and BT1285 (green). Assays were run in technical triplicates (n = 3). Data are presented as mean values ± SD. Source data are provided as a Source Data file.

BT1285. Curiously, both HM processing ENGases, Alfi_0894 and Alfi_0882, exhibited a high binding affinity to Man9-IgG (Supplementary Fig. 18). Thus, while *A. finegoldii* does not replicate *B. thetaiotaomicron* VPI-5482 identically in terms of its HM-processing

endoglycosidases, this bacterium does unequivocally encode two endoglycosidases with distinct functional properties that depend on both the environmental conditions in which it functions as well as the conformational accessibility of the HM *N*-glycan substrate.

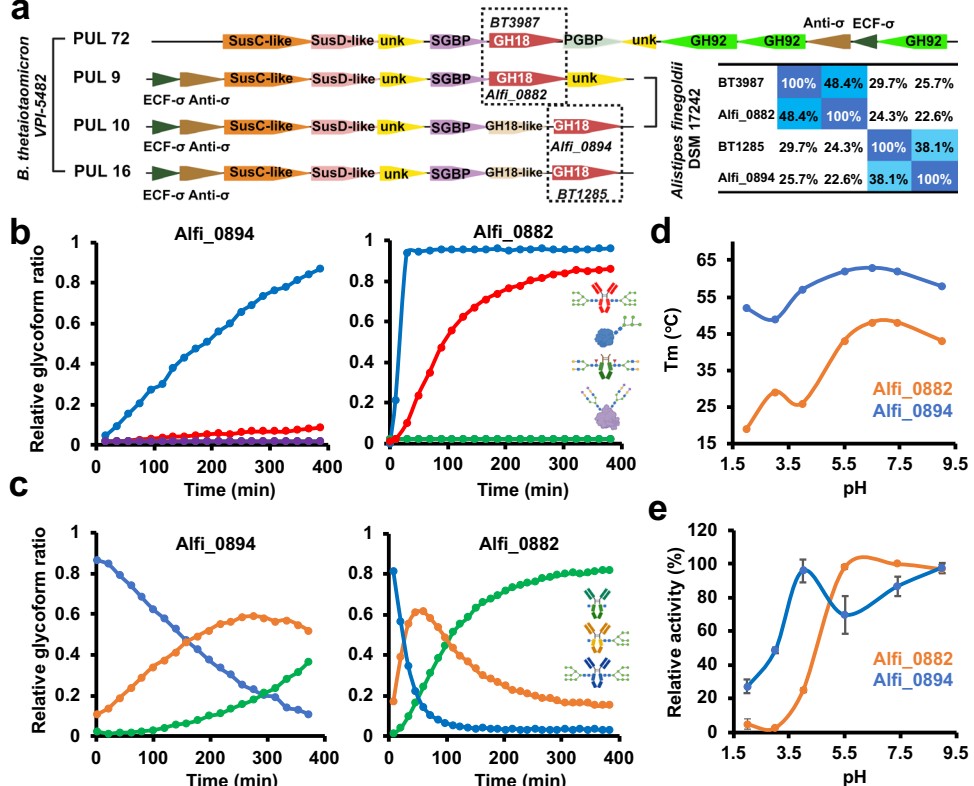

**Fig. 7 | Two different HM-ENGases conservation in *Alistipes finegoldii*.**
**a** Schematic representation of putative High-Mannose PULs in *A. finegoldii* in comparison with High-Mannose PULs in *B. thetaiotaomicron*. SGBP=surface glycan binding protein, PGBP=peptidoglycan binding protein, unk=unknown function, ECF=extracellular factor. Left pannel shows sequence identity percentages of *B.thetaiotaoimicron VPI–5482* and *A. finegoldii* HM-ENGases. **b** Competitive kinetic analysis of *Alfi*_0894 and *Alfi*_0882. Curves represent the complete deglycosylated substrate proportion. **c** Kinetic analysis of *Alfi*_0894 and *Alfi*_0882. Representative of technical triplicates (*n* = 3). **d** Tm vs pH curves of both HM-ENGases from *A. finegoldii* obtained by DSF analysis. **e** Relative ENGase activity at different pHs, of both HM-ENGases from *A. finegoldii* obtained by LC-MS analysis using RNAseB as a substrate at 20 °C. Assays were run in technical triplicates (*n* = 3). Data are presented as mean values ± SD. Sugar symbols: ■ GlcNAc, ● Man, ▲ Fuc, ● Gal, ◆ Neu5A. Source data are provided as a Source Data file.

## Discussion

In this study, we have shown that a bona fide GH18 ENGase, BT1285, is involved in HM *N*-glycan catabolism and is located in a different PUL than is BT3987 in the genome of the human gut symbiont *B. thetaiotaomicron* VPI-5482. BT1285 is a 30 kDa enzyme composed of a single GH18 catalytic domain with a TIM barrel fold typical of ENGases, while BT3987 contains an additional β-sandwich domain in the N-terminal region (DUF1735). BT1285 and BT3987 share less than 30% sequence identity and populate distinct clusters in our SSN data (Fig. 1a), accompanied by an RMSD of 2.4 Å between both ENGases, indicative of structural divergence and suggestive of the functional differences that we observe here. The genome of *B. thetaiotaomicron* contains the largest number of extracytoplasmic function (ECF) σ-factors among bacteria and archaeal species[59]. These ECF σ-factors are able to sense changes in the environment and act as essential dissociable protein subunits of prokaryotic RNA polymerase necessary for initiation of transcription. ECF σ-factors and their membrane-tethered cognate anti σ-factors are mostly located upstream of PULs, suggesting a finely tuned regulatory system that allows *B. thetaiotaomicron* to sense the nutrients and adjust its metabolism accordingly, thus benefiting itself and the human host[59]. Most ECF σ/anti σ-factor in *B. thetaiotaomicron* seem to be mainly associated with PULs involved in the specific catabolism of host glycans[30].

Previously, a sole PUL, PUL72 containing BT3987, involved in the processing of HM-*N*-glycans in *B. thetaiotaomicron*, was described on the basis of transcriptional analysis of SusC genes of only PULs containing GHs belonging to the GH92 family, which are required for

processing mono- and disaccharides containing mannose sugars; PUL72 was indicated to be the only PUL induced in cultures grown in the presence of Man$_8$ *N*-glycans[37]. However, a knock-out mutant of the PUL72 ECF σ-factor BT3993 was capable of growth, albeit after a substantial lag phase, in media where the sole carbon source was Man$_8$ *N*-glycan, which could be explained by the eventual induction of PUL16-containing BT1285. Additionally, PUL72 and PUL16 each contain their own ECF σ-factor/anti σ-factor pairs (BT1278/BT1279 and BT3993/BT3992, respectively, Fig. 1b), indicating that the expression of both of the HM processing endoglycosidases, BT3987 and BT1285, is not co-regulated in *B. thetaiotaomicron* and is likely independently adjusted to adapt to alterations in the environment.

Furthermore, the analysis of transcriptional profiling (RNA-seq) from *B. thetaiotaomicron* grown in vitro and in vivo in different conditions (available in GEO profiles in the NCBI database, see methods)[26,27] showed that in chemostat cultures both BT1285 and BT3987 were highly expressed in rich media (TYG) and low or no expression was detected in minimal medium containing glucose or maltotriose at any stage of the growth phase (Supplementary Fig. 19). BT1285 reached high transcription levels at stationary phase when *B. thetaiotaomicron* was grown in rich medium, whereas BT3987 reached high expression in *B. thetaiotaomicron* grown in the ceca of colonized mice on polysaccharide or simple sugar diets (Supplementary Fig. 19). Hence, this RNA-seq data indicated that the two HM PULs, PUL72 and PUL16, are expressed in different environmental conditions both in vitro and in vivo. In addition, another transcriptomic analysis indicated that PULs 16 (BT1280-85) and 72 (BT3983-88) of *B.*

*thetaiotaoimicron* seemed to be responsive to mucin *O*-glycan fractions in vitro[30]. It was also reported that MUC1 mucin contains high-mannose structures (Hex5-9HexNAc2) and complex/hybrid-type glycans (NeuAc0-3Fuc0-3Hex3-8HexNAc3-7) in addition to *O*-glycans[28,29].

In *B. thetaiotaomicron* several PULs have been shown to target polysaccharides rich in α-linked mannose sugars[37], which encode numerous genes belonging to α-mannoside-specific CAZy families, including GH38, GH76, GH92 and GH125, and result in apparent redundancy[60]. The ability to utilize a particular glycan substrate activates an extensive enzymatic apparatus encoded by multiple loci. For example, the breakdown of CT *N*-glycans by *B. thetaiotaomicron* VPI-5482 requires the cooperative action of a complex multi-locus-encoded molecular machine containing multiple ENGases[11], which may exhibit different functionality, similar to those we describe here for HM-processing ENGases. In this context, PUL72 containing BT3987 encodes three periplasmic α-mannosidases from the GH92 family to hydrolyse the oligomannose into a trisaccharide that is degraded by enzymes that are not encoded by this HM *N*-glycan PUL[37], whereas PUL16 containing BT1285 does not encode any mannosidases, suggesting that HM *N*-glycans imported by PUL16 are processed to completion by mannosidases encoded in yet a different PUL.

The final metabolites of *N*-glycan catabolism released by Bacteroidales are short chain-fatty acids (SCFAs) including acetate(C2) and propionate(C3), which are produced by fermentation of monosaccharides and primarily absorbed and used by the host as a key substrate for energy production[61]. SCFAs contribute to the maintenance of intestinal barrier integrity through mucus production and increased expression of tight junction proteins. SCFAs are detected mainly in the colon, but also present in the liver and blood[61]. SCFAs can also inhibit the invasion and colonization of pathogens by lowering the intestinal pH and also have various anti-inflammatory properties[62]. Despite the pH in the human gut environment typically ranging from 5.5 to 7.5[63], a clinical study found that 50% of patients with active ulcerative colitis showed a high acidic proximal colonic pH (ranging between pH 2.3 and 3.4)[64]. The gut microbiota is sensitive to pH levels, and changes to this environmental factor can result in large shifts to community structure and function[65,66]. A high level of SCFAs produced by glycan fermentation may temporarily reduce the local pH of the gut extracellular environment which could regulate the activity of secreted ENGases. We associate the broad spectrum of optimal pH of BT1285 to a higher degree of protein flexibility and an ability to access largely inaccessible glycans, relative to BT3987. Thus, the catalytic activity of BT1285 and BT3987 processing HM glycoprotein substrates from the host and/or food sources could play a role in controlling some inflammatory processes in different conditions in the human gut environment, through the generation of SCFAs and/or direct through removing HM glycans on relevant human glycoproteins such as immunoglobulins, Fc γ Receptors and complement components, etc. Moreover, it has been reported that *Ef*Endo18A, one of the two GH18s encoded by *Enterococcus faecalis*[67,68], is up-regulated in blood and urine[69,70], where *E. faecalis* frequently causes infection. It was proposed that this ENGase can inactivate host glycoproteins to provide an advantage to the bacteria during infection. Considering that BT1285 is highly similar to *Ef*Endo18A (Supplementary Fig. 7a, b), and that in some rare cases Bacteroidales can colonize extraintestinal sites[59,71,72], we cannot rule out that the alternate expression of BT1285 may help the bacteria to process host glycoproteins to support bacterial infection.

We also showed that in addition to *B. thetaiotaomicron*, another relevant human gut bacteria, *Alistipes finegoldii*[58,73,74] can produce two GH18 ENGases with the same HM *N*-glycan substrate specificity but with distinct functional properties (Fig. 7), an apparent conserved phenomenon widespread amongst bacterial species that prominently colonize the human gut (Supplementary Fig. 17), some of which are currently being used as probiotics or postbiotics (e.g., *B.*

*xylanisolvens*[75,76] and *B. ovatus*[77–79]). Moreover, it was recently proposed that carbohydrate metabolism by Bacteroidales (mainly *A. indistinctus*, *A.finegoldii* and *B. thetaiotamicron*) has a potential therapeutic role in ameliorating diet-induced obesity and insulin resistance[74].

Thus, we identified at least two functionally distinct endoglycosidases involved in the specific processing of HM *N*-glycans, encoded within separate PULs in different gut bacteria. These ENGases exhibited optimal stabilities and activities in different environmental conditions when processing their *N*-glycan substrates on two different glycoproteins. Together, our findings indicate that human gut microbes commonly express multiple endoglycosidases with the same substrate specificity and suggest that these enzymes are not simply redundant but are instead part of an evolutionary strategy employed by these bacteria to survive and thrive throughout the range of environmental conditions encountered in the human gastriointestinal tract. Moreover, inactive mutants of ENGases generated in this work (e.g. BT1285i) that recognize different HM *N*-glycans with such high affinity could potentially be used as artificial lectins with high specificity for targeting tumor-associated HM *N*-glycans molecules, potentially aiding cancer treatment. Additionally, such artificial lectins could be used to purify therapeutic glycoproteins, such as antibodies, which often have some contaminating fraction of HM *N*-linked glycoforms, for biomedical applications.

## Methods
### Bioinformatic tools
We constructed sequence similarity networks (SSN) using all annotated GH18 enzymes from the CAZY database (www.cazy.org)[41], and the Enzyme Similarity Tool (EFI-EST)[40] (https://efi.igb.illinois.edu/), selecting an alignment score threshold of 50 (in order to reach an alignment stringency that allowed separation into distinct clusters in which proteins shared sequence identity lower than 35%). Then, we used Cytoscape software (https://cytoscape.org/)[80] for deep analysis of the resulting SSNs. An analysis of the composition of each cluster is presented in Supplementary Data 1. SSN analysis was made using the full-length sequence of GH18 family proteins, however, this analysis is based on a local alignment that generate cluster based on similarity of protein domains. Uniprot database (https://www.uniprot.org/)[81] and AlphaFold (AF) (https://alphafold.com/)[42] were used for extracting DNA and protein sequences and protein folding analysis and ncbi (GEO profiles) (https://www.ncbi.nlm.nih.gov/geoprofiles/). https://www.ncbi.nlm.nih.gov/geo/tools/profileGraph.cgi?ID = GDS1849:BT1285_at and https://www.ncbi.nlm.nih.gov/geo/tools/profileGraph.cgi?ID = GDS1849:BT3987_at). Protein BLAST against PDB database (https://www.rcsb.org/), predicted AF model and presence of typical β-hairpin around active site was used to identify HM-specific ENGases. JGI Integrated Microbial Genomes (IMG) database (http://img.jgi.doe.gov) and PULDB were used to analyze polysaccharide utilization loci (PUL) from Bacteroidales. PUL modular alignment analysis tools (in PULDB)[17] was used for searching conserved HM PULs in Bacteroidales. Schemes/Diagrams were created with BioRender.com (under a full Academic licence) and Inkscape v.1.3.2.

### Cloning, expression, and purification of bacterial proteins
The pETSUMO (pET28a containing SUMO tag) vector encoding the BT1284 and BT1285 protein (*B. thetaiotaomicron* VPI-5482) was constructed by amplifying genomic DNA (purchased from ATCC (https://www.atcc.org/products/29148d-5.) with specific primers (Supplementary Table 1 and 2). Single and double-point mutations were designed using the NebBaseChanger (https://nebasechanger.neb.com/) and PCR was carried out using the NEB Q5® High Fidelity polymerase and manufacturer's instructions. Full sequences were confirmed by Genwiz (https://www.genewiz.com). pET28a containing Alfi_0882 and Alfi_0894 genes were ordered to Twist Bioscience (San Francisco,

CA, USA). *Escherichia coli* BL21(DE3) cells (NEB) transformed with the corresponding plasmid were grown in 2000 ml of LB medium supplemented with 50 µg ml−1 kanamycin at 37 °C. When the culture reached an $OD_{600}$ value of 0.6–0.8, the expression of proteins was induced by adding 0.2 mM Isopropyl β-d-1-thiogalactopyranoside (IPTG). After ca. 16 h at 18 °C, the cells were harvested by centrifugation at 6000 × g for 20 min at 4 °C and resuspended in 50 ml of 20 mM HEPES, pH 7.4, 300 mM NaCl. Cells were disrupted by sonication (12 cycles of 10 s pulses with 60 s cooling intervals between the pulses, and 60% of amplitude) at 4 °C, and the suspension was centrifuged at 10,000 × g for 10 min at 4 °C. The supernatant after being filtrated by 0.2 µm and then applied into a Nickel-Nitrilotriacetic acid (Ni-NTA) column (1 ml, Thermo Scientific) equilibrated with 20 mM Hepes, pH 7.5, 300 mM NaCl. The elution was performed with a linear gradient of 0 to 500 mM imidazole in 20 ml of solution A at 1 ml min−1. The eluted fractions were buffer exchanged to 20 mM HEPES pH 7.4, 150 mM NaCl, pH 7.4 in an Amicon Ultra-15 centrifugal filter unit (Millipore) with a molecular cutoff of 10 kDa at 4000 × g., and then incubated with enzymatic digestion with ULP-1 protease (SUMO protease) overnight at 4 °C (or 4 h at RT). Mixture was applied again into a Ni-NTA column (1 ml, Thermo Scientific) and flow through was recovered. Proteins were further purified by size-exclusion chromatography using a Superdex 200 10/300 GL column (GE Healthcare) equilibrated in 20 mM HEPES pH 7.4, 150 mM NaCl, pH 7.4. The completeness of the enzymatic digestion with SUMO protease reaction was confirmed by SDS-PAGE. The eluted protein was concentrated using an Amicon Ultra-15 centrifugal filter unit (Millipore) with a molecular cutoff of 10 kDa at 4000 × g.

### Crystallization and data collection

BT1285 wt was crystallized by mixing 0.5 µL of a protein solution at 20 mg mL−1 in 20 mM Buffer Phosphate pH 7.4, 150 mM NaCl with 0.5 µL of Index 19 (0.056 M Sodium phosphate monobasic monohydrate, 1.344 M Potassium phosphate dibasic, pH 8.2). Crystals grew in 3-5 days. Crystals were quickly immersed in mother liquor containing 500 mM NaI and 20% (vol/vol) ethylene glycol before flash freezing[82]. A complete X-ray diffraction dataset was collected at Beamline Manacá (Sirius, LNLS-CNPEM, Campinas, Brazil). BT1285 apo +NaI structure crystallized in the P 21 21 2 space group with one molecule in the asymmetric unit and diffracted to a maximum resolution of 1.08 Å (Supplementary Table 3). In addition, BT1285 apo and $BT1285_{D161A-E163A}$ were also crystallized by mixing 0.5 µL of a protein solution at 80 mg mL−1 in 20 mM Buffer Phosphate pH 7.4, 150 mM NaCl with 0.5 µL of Morpheus A8 (Molecular Dimensions) (12.5% w/v PEG 1000, 12.5% w/v PEG 3350, 12.5% v/v MPD 0.03 M of each divalent cation 0.1 M MOPS/HEPES-Na pH 7.5). Crystals were transferred to a cryo-protectant solution containing 20% glycerol and frozen under liquid nitrogen. Complete X-ray diffraction datasets were collected at the SER-CAT Synchrotron beamline 22-ID (USA). Both, BT1285 and $BT1285_{D161A-E163}$ structure crystallized in the P 21 21 2 space group with one molecule in the asymmetric unit and diffracted to a maximum resolution of 1.28 Å and 2.10 Å (Supplementary Table 3). The $BT1285_{D161A-E163A}$-$Man_9GlcNAc_2$ complex was crystallized by mixing 0.8 µL of a protein solution at 80 mg ml−1 in 20 mM Hepes pH 7.4, 150 mM NaCl and 7 mM Man9GlcNAc2Asn, with 0.5 µL of Index 19 (Hampton Research) (0.056 M Sodium phosphate monobasic monohydrate, 1.344 M Potassium phosphate dibasic, pH 8.2) and 0.2 µL of seed stock of crystals obtained in Index 5 condition (0.1 M HEPES pH 7.5, 2.0 M Ammonium sulfate). They were transferred to a cryo-protectant solution containing 20% glycerol and frozen under liquid nitrogen. Complete X-ray diffraction datasets for crystal forms were collected at beamline 22-ID (SER-CAT). $BT1285_{D161A-E163A}$-$Man_9GlcNAc_2$ complex crystallized in the primitive monoclinic space group P 1 21 1 with two molecules in the asymmetric unit and diffracted to a maximum resolution of 1.9 Å

(Supplementary Table 3). Datasets were integrated and scaled with HKL2000[83] following standard procedures. The HM glycan $Man_9GlcNAc_2$ used for co-crystallization assays was purchased from NatGlycan (Atlanta, GA). We obtained two *B. faecium* GH18-like structures that crystallized in the C 2 1 and P 1 21 1 space groups with one and two molecules in the asymmetric unit and diffracted to a maximum resolution of 2.67 Å and 2.90 Å, respectively (Supplementary Table 4). The *B. faecium* GH18-like crystallized by mixing 0.6 µL of a protein solution at 80 mg ml−1 in 20 mM Hepes pH 7.4, 150 mM NaCl with 0.4 µL of 0.2 M Sodium bromide, 0.1 M Bis Tris propane pH 7.5, 20% (w/v) PEG 3350 (PACT 2-26(G2)) and 0.2 µL of seed stock of crystals obtained (4× diluted) in PACT 2-26(G2) condition. Crystals were transferred to a cryo-protectant solution containing 20% glycerol and frozen under liquid nitrogen. We incubated protein solution with 3.5 mM Man9GlcNAc2Asn for co-crystallization assays but we did not observe any visible density of the glycan in the crystal data. Complete X-ray diffraction datasets for crystal forms were collected at beamline 22-BM (SER-CAT).

### Structures determination and refinement

The BT1285 WT apo structure was initially determined through SAD, employing iodine as the anomalous scattering source[82]. The heavy atom substructure was determined using SHELXD[84], and the structure was subsequently phased with PHASER[85]. The initial model was automatically built in PHENIX, and iterative rounds of rebuild and refinement conducted in COOT[86] and PHENIX[87] yielded the final structure. Structure determination of BT1285 WT without sodium iodine, BT1285 D161A-E163 (BT1285i) and BT1285-$Man_9GlcNAc_2$ complex was carried out by molecular replacement methods implemented in Phaser and the PHENIX suite[87] using the crystal structure of BT1285 as a template model. The manual building of the glycan was performed with Coot and refinement with phenix.refine. Structure determination of *B. faecium* GH18-like (Uniprot ID: A0A6H0KVH8) was carried out by molecular replacement methods using the AlphaFold modeling of *B. faecium* GH18-like as a template model. The structures were validated by MolProbity[88]. Data collection and refinement statistics are presented in Supplementary Tables 3 and 4. Molecular graphics and structural analyses were performed with Pymol (PyMOL Molecular Graphics System, Version 1.2r3pre, Schrödinger, LLC) and UCSF Chimera 1.16[89].

### Purification of Fc and IgG1

IgG1 (Rituximab) plasmid and Fc-IgG1 were expressed in Expi293 cells (ATCC) and/or HEK cells using polyethyleneimine as a transfection agent. Kifunensine, a potent inhibitor of the mannosidase I enzyme, was used to ensure HM glycans were present on the Fc and IgG1. Fc-IgG1 containing a mixture of G0F, G1F and G2F (defined as complex type *N*-glycans or CT) were obtained by culture cells not treated with Kifunensine. After transfection, cells were cultured for 96 h in Free-style F17 medium supplemented with GlutaMAX and Geneticin (Thermo Fisher Scientific). HM IgG1 was purified using protein A chromatography, with 20 mM sodium phosphate buffer pH 7.0 used as a binding buffer and 100 mM sodium citrate buffer pH 3.0 as elution buffer. The fractions were neutralized with 1 M Tris pH 9.0. SDS-PAGE was used to identify fractions which contained HM IgG1. These were subsequently pooled, concentrated and analyzed by LC-MS. Monoglycosylated IgGs used for SPR assays were obtained using a previously established hetereodimeric platform with mutations L368D/K370S on one chain (R) and E357Q/S364K on the other (L)[90]. The glycosylation site at N297 were removed from either chain via a N297Q mutation. Aglycosylated IgG was produced using the heterodimeric platform with both chains containing the N297Q mutation. IgG-$Man_5$ was produced by incubating IgG-$Man_9$ with alpha α (1-2)-Mannosidase (*Aspergillus saitoi*) at room temperature.

## Hydrolytic activity assays

LC-MS kinetic analysis was used to determine the reaction rate of deglycosylation by BT1285, BT3987, Alfi_0882, Alfi_0894. A total of 30 μL reactions were set up containing 2 μM of the Fc variant and 50 nM of the enzymes in HBS (20 mM HEPES pH 7.4; 150 mM NaCl). The reactions were analyzed by LC-MS using an Agilent 1290 Infinity II LC System equipped with a 50 mm PLRP-S column from Agilent with 1000 Å pore size. The LC system is attached to either an Agilent 6545XT quadrupole-time of flight (Q-TOF) (Agilent, Santa Clara, CA). The reactions were setup and placed in the LC-MS and the reactions were sampled approximately every 6.5 min until completion. All reactions were performed in triplicate. Relative amounts of the substrate and hydrolysis products were quantified after deconvolution of the raw data and identification of the corresponding peaks using BioConfirm10.0 (Agilent, Santa Clara, CA). For activity mesearuments at different pHs, 20 μl aliquots of the reaction (50 nM enzyme mixed with 2 μM RNAseB in HBS pH 7.4 at 23 C were taken in triplicate and allowed to progress for 30-60 min. We also analyzed the catalytic activity of BT1285, BT3987, BT1284 and *B. faecium* GH18-like against plant complex-type *N*-glycans using Horseradish peroxidase (HRP) as a substrate. HRP was purchased from Sigma (77332). We determined activity of these proteins by incubating with HRP, IgG CT or RNAseB 24 hs at 37 °C in buffer HBS pH 7.4 and then loading the samples in SDS-PAGE gels.

## Differential scanning fluorimetry (DSF) or Thermal shift assay (TSA)

BT1285, BT3987, Alfi_0882, Alfi_0894, and GH18 BT3987 proteins were diluted to reach a concentration of 5–10 μM protein with 25× Sypro-Orange dye (ThermoFisher) (in the dark) into the assay buffer in 20 μl final volume. The buffer used were 50 mM pH 2.0 (Hydrochloric Acid-Potassium Chloride Buffer), 50 mM pH 3.0 (citrate buffer), 50 mM pH4.0(Acetate buffer), 50 mM pH 5.5 and 6.5 MES buffer, 50 mM pH 7.4 Hepes, and 50 mM pH 9.0 Tris-HCl with 150 mM NaCl. The 20 μl samples were loaded into a 96-well qPCR plate and incubated into a Real-Time thermocycler (Applied Biosystems), then the assay plate was briefly centrifugated ( ~1000 × g, 1 min) to mix compounds into the protein solution, and the plates were sealed with a transparent plastic film. A temperature ramp rate of 1 °C/min was used to measure the melting temperatures (Tm) over a temperature rate of 15–95 °C. The Tm curves of each enzyme variant were analyzed and calculated in each condition using Excel.

## Surface plasmon resonance (SPR)

SPR experiments were performed by using a Biacore X100 instrument (GE Healthcare) from the Emory Glycomics and Molecular Interactions Core (EGMIC) at Emory University. For SPR experiments we used CM5 and proteinA sensor chip (GE Healthcare). For CM5 sensor chip, we immobilized deglycosylated protein (RNAseB) in Flow cell 1 and the glycosylated variant (RNAseB Man5/6) was immobilized on Flow Cell 2. N-linked glycan of RNAseB in flow cell 1 was removed using 3 × 20 μL of 1 mg/mL BT1285 wt and this flow cell was used as the negative control surface. RNAseB was inmmobilized by standard amine-coupling procedure in 10 mM sodium acetate, pH 4.5 at a density of 150 RU. Concentration series of all BT1285i, BT1285i$_{H126A}$ or BT3987i proteins were injected and kinetics and/or affinity constants were calculated where possible. Serial dilutions of proteins were injected in HBS-EP running buffer (10 mM HEPES, 150 mM NaCl, 0.005 % Tween 20, EDTA 3 mM) over flow cells 1 and 2 for 120 s per injection and allowed to dissociate for 600 s. Between binding cycles, the sensor chip surface was regenerated by washing with with one cycle each of of 2 M NaCl and 10 mM HCl. For Protein A chip, injection of IgG1 HM (Man$_5$/Man$_5$, Man$_9$/Man$_9$ or Man$_9$) on Fc2 (up to 400 RU, 15 nM of IgG1) and then injection of serial dilutions of inactive enzyme as analyte. ProteinA chip regeneration was performed by one step of 10 mM HCl in each cycle.

Affinity constants for BT1285i $_{H126A}$ were calculated using a general steady-state equilibrium model and kinetics for BT1285i and BT3987i were fitted with the Biacore X100 evaluation software.

## Analytic ultracentrifugation and SEC-MALS

Sedimentation velocity analytical ultracentrifugation was conducted using an An-50 Ti analytical rotor at 50,000 rpm (182,000 × g) at a nominal temperature of 20 °C in a Beckman Coulter XLI analytical ultracentrifuge using standard procedures[91]. BT1285i was incubated with Fc HM mixture at a nominal ratio of 2:1 (70 μM BT1285i- 35 μM Fc dimer) and SEC-purified BT1285/Fc HM fraction, BT1285i and FcHM were run individually. The partial specific volumes of BT1285i, FcHM and BT1285i-FcHM were calculated based on amino acid composition using SEDFIT version 16.36 (https://spsrch.cit.nih.gov/). Samples (0.4 mL) were loaded into 12 mm pathlength Epon double sector cells equipped with sapphire windows with matched buffer (20 mM Hepes 150 mM NaCl, pH 7.4) in the reference sector. The buffer density and viscosity at 20 °C were measured using an Anton Parr DM4500 densitometer and Lovis 2000M viscometer. Absorbance scans at 280 nm were initiated after reaching the target rotor speed and collected at 4.7 min intervals. Data were corrected for scan time errors using REDATE version 1.01[92]. Data were analyzed using the continuous $c(s)$ distribution model[93] in SEDFIT and a sedimentation coefficient interval of 0 to 10 S at 0.1 S intervals. Data were fitted using sequential simplex and Marquardt-Levenberg algorithms and maximum entropy regularization with a confidence interval of 0.68. The fitted parameters were $c(s)$, the frictional ratio ($f/f_0$), time-invariant noise and the meniscus position. Sedimentation coefficients were adjusted to the standard condition of 20 °C in solvent water. Molecular weight estimates were obtained for the dominant species in buffer-exchanged Advate and Kogenate from the Svedberg equation by deconvoluting the contribution of diffusion to the observed signal as described[93].

The SEC MALS configuration consisted an isocratic pump/vial sampler/variable wavelength 1 cm pathlength detector (1260 Infinity II HPLC system, Agilent Corporation) in line with a Superdex 200 10/300 GL SEC column (GE Healthcare Life Sciences), DAWN multi-angle light scattering detector and Optilab differential refractometer (Wyatt Corporation). A 0.1 mL mixture of 2.1 mg/mL BT1285 and 1.9 mg/mL Fc-HM was applied to the SEC column at 0.5 mL/min at room temperature. HPLC control, data acquisition and analysis were performed using ASTRA (version 8.1.2.1) and HPLC Manager (version 1.4.1.1) (Wyatt Corporation). In aqueous solvents, DAWN measures scattering of vertically polarized 658.3 nm GeAs laser light at 17 angles ranging from 28 to 147 degrees. Voltage signals from 0.5 s "slices" of the chromatogram were converted to Rayleigh ratios, $R_\theta$, by calibrating 90 degree light scattering with toluene according to instructions provided by the manufacturer. Normalization of the scattering at the other angles to the signal at 90 degrees was done using bovine serum albumin as an isotropic scatterer. Normalization, peak alignment and correction for band broadening were performed using ASTRA. Estimates of weight average molecular weight, $M_w$ were made in ASTRA using the Zimm model.

## SEC-SAXS experiments

A 240 μM solution of BT1285i was incubated with 80 μM solution of Fc-IgG HM in 25 mM Hepes pH 7.0, 150 mM NaCl, 0.5 mM TCEP, for 5 min at room temperature to form the BT1285-Fc-IgG1(HM) complex. Small-Angle X-ray Scattering coupled with Size Exclusion Chromatography (SEC-SAXS) data for purified BT1285i, and Fc-IgG1 and the complex mixtures of BT1285i-FcIgG1(HM) were collected on the ID7A1 beamline for Bio SAXS of MacCHESS (Macromolecular X-ray science at the Cornell High Energy Synchrotron Source, Cornell University, U.S.A). 30 μL of the protein samples were injected into a 2.4 ml Superdex200 Increase 5/150 and the complex into a 24 ml Superdex200

Increase 10/300 and eluted at a flow rate of 0.2 ml/min at 4 °C. Bio-SAXS experiments were run at 11.3 keV. Data were collected using a EigerX 4 M detector (Dectris, CH) at a sample-detector distance of 1722 mm and a wavelength of λ = 1.102 Å. The range of momentum transfer (s) of 0.0098 to 0.48 nm$^{-1}$ was covered (s = 4πsinθ/λ, where θ is the scattering angle). Data was processed and merged using standard procedures by the program package BioXTAS RAW[79,94–96] and ATSAS[97]. The maximum dimensions ($D_{max}$), the interatomic distance distribution functions ($P(r)$), and the radii of gyration ($Rg$) were computed using GNOM. The ab initio multiphase reconstruction of the SAXS data of the BT1285i-Fc-IgG1 complex was generated using the GASBOR[98] algorithm from ATSAS[97]. The results are summarized in Supplementary Table 5.

## Negative staining TEM

Inspection of the BT1285$_{D161A-E163A}$-Fc-HM complex was performed by negative stain transmission electron microscopy (NS-TEM). An aliquot of 50 μL of the complex BT1285$_{D161A-E163A}$-Fc-HM (1:3) were injected into an analytical size exclusion chromatography column (Shodex KW403-4F) using PBS pH 7.4, as running buffer. The single peak corresponding to the complex MW was collected in 30 μL fractions for subsequent standard negative stain grid preparation using continuous glow discharged carbon electron microscopy grids CF400-CU (EMS, USA). Briefly, drops of 3 μL of diluted complex sample were placed on the grid, incubated for 90 s and then blotted with filter paper (Whatman®) to remove the excess. The grids were washed by placing them onto a drop of PBS, pH 7.4 for 30 s and blotted dry. The complex was stained by transferring the grid to a drop of 1% w/v uranyl formate for 60 s, then dried with a filter paper and stored at room temperature. Samples were imaged at 120,000× magnification using a Talos L120C 120 kV D5550 CryoTwin equipped with an BM-Ceta camera (Gatan, USA) from the Robert P. Apkarian Integrated Electron Microscopy Core at Emory University, GA Ca. 300 images collected were used to analyze particles by single particle analysis using Cryo-SPARC and for obtaining a low resolution reconstruction.

## Reporting summary

Further information on research design is available in the Nature Portfolio Reporting Summary linked to this article.

## Data availability

The atomic coordinates and structure factors have been deposited with the Protein Data Bank, accession codes 8U9F (BT1285 wt+NaI), 8U46 (BT1285 D161A-E163A), 8U47 (BT1285 wt), 8U48 (BT1285 + Man9GlcNAc2), 8W01 (*B. faecium* GH18-like C2) and 8W04 (*B. faecium* GH18-like P21). Previously reported PDB structures used in this study are available under the accession codes: 8UWV, 6TCV, 2WVX, 4MRU, 5JII, 7NWF, 1C8Y, 6Q64, and 3POH. Other data are available in the Supplementary information and Supplemenatary Data. Source data are provided with this paper.

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

## Acknowledgements

This work was supported in part by NIH R01GM148075 (to E.J.S.). We would like to thank SER-CAT APS (USA) Beamline BM-22 and ID-22 APS and SIRIUS LNLS-CNPEM Brazilian Synchrotron, Beamline Manacá. We would like to thank BioSAXS beamline (ID7A1) at CHESS, especially Qingqiu Huang and Richard Gillilan for collecting SEC-SAXS data. We thank Prof. Lai-Xi for providing α-1-2-mannosidase. We thank Humberto D'Muniz Pereira from IFSC-USP for helping in the collection of X-ray diffraction data of structure 8U9F. We thank Prof. Caterina G.C.M. Netto from UFSCar-Brazil for the careful reading and insightful feedback on the manuscript. We thank Agilent Technologies for supplying a 1290 Infinity II LC system and a 6545XT Bio LC/QTOF for use in the Sundberg laboratory.

## Author contributions

D.E.S. and E.J.S. conceived the project. D.E.S., N.S., M.N., M.H., J.D., J.O.C., M.F., X.L., and P.L. performed the experiments, D.E.S., N.S., M.N., M.H., J.D., J.O.C., X.L., P.L., B.T., M.E.G., and E.J.S. analyzed the results. D.E.S. and E.J.S. wrote the paper.

## Competing interests

Based on the development of an artificial lectin, BT1285i, with high specificity and high affinity for oligomannose *N*-glycans, D.E.S. and E.J.S. are inventors on a provisional patent application No. 63/538,956 filed with the Patent and Trademark Office by Emory University. All other authors declare they have no competing interests.

## Additional information

[1]Department of Biochemistry, Emory University School of Medicine, Atlanta, GA, USA. [2]Institute of Physics (IFSC-USP), University of São Paulo, São Carlos, SP, Brazil. [3]Instituto Biofisika (UPV/EHU, CSIC), University of the Basque Country, Leioa, Spain. [4]Department of Pediatrics, Emory University School of Medicine, Atlanta, GA, USA. [5]Structural Glycoimmunology Laboratory, Biobizkaia Health Research Institute, Barakaldo, Bizkaia, Spain. [6]Ikerbasque, Basque Foundation for Science, Bilbao, Spain. [7]Structural Glycobiology Laboratory, Department of Structural and Molecular Biology, Molecular Biology Institute of Barcelona (IBMB), Spanish National Research Council (CSIC), Barcelona Science Park, c/Baldiri Reixac 4-8, Tower R, Barcelona, Catalonia, Spain. [8]Present address: Structural Biochemistry Unit, National Institute of Dental and Craniofacial Research (NIDCR/NIH), Bethesda, MD, USA. [9]Present address: Center for Innovative Proteomics, Cornell University, Ithaca, NY, USA. [10]Present address: Sydney Pharmacy School, Faculty of Medicine and Health, The University of Sydney, Camperdown, NSW, Australia. ✉e-mail: dsastre@emory.edu; eric.sundberg@emory.edu

