## [Peer Review File · Nature Communications]

Reviewers' Comments:

Reviewer #1:

Remarks to the Author:

This paper by Sastre et al. presented biochemical, biophysical, and structural characterizations of BT1285. BT1285 is shown with ample evidence to be a second HM-processing ENGase (endo- β -N-acetylglucosaminidase) from the GH18 family in *B. thetaiotaomicron*, acting on high-mannose (HM) N-glycans in glycoproteins. They also compared the molecular mechanisms of the two ENGases (BT1285 and BT3987), and identified HM-processing ENGases in other gut bacteria. The functional characterizations to me seem very comprehensive and supporting the conclusions. I have some comments on the methods and presentations:

1. Fig 1: authors may add a diagram to show the different Pfam domain architectures of the 4 highlighted SSN clusters (e.g., DUF1735 in BT3987/C2 and DUF4849 in C1/C4). Maybe select one representative protein from each cluster and draw their domain start and end positions and the total protein lengths.
2. Fig 1a: I would like to see EC numbers labeled beside SSN clusters if available. Are there other clusters with ENGase (EC 3.2.1.96) activities beside C2 and C3? Do C2 and C4 have known activities?
3. CAZy GH18 page shows 53,101 proteins, while line 135 mentioned 36,188 were used for SSN analysis. This might be because the data were downloaded a few years ago. Please supply the time when you downloaded GH18 data from CAZy. Could you provide reasons why alignment score threshold 50 was used?
4. Paragraph in lines 445-463: some method details are needed to understand how the bioinformatics search in Bacteroidetes phylum was done. It seems to me that the search was limited to PULs in PULDB. Did you use BT1285 and BT3987 proteins to blast against PULs? Would it be possible that some better homologs exist in *Alistipes finegoldii* but not in PULs? For that, you could expand the search to the entire genomes of *Alistipes finegoldii* instead of only its PULs.
5. Line 530 and Figure S16: provide GEO dataset accessions that are used to make the expression plots.
6. Line 587, Figure S13 -> S14?
7. Line 590, PULS -> PULs.
8. Line 615, check the sentence punctuation.

Reviewer #2:

Remarks to the Author:

This manuscript by Sastre et al. presents the biochemical and structural characterisation of Glycoside Hydrolase Family 18 (GH18) N-glycan endo-glycosidases from human gut bacteria, primarily focussing on two paralogues from *Bacteroides thetaiotaomicron*. Biochemical characterisation of a pair of orthologues from *Alistipes finegoldii* is also included. Briefly, a range of classical biochemical data was presented, including pH-activity and thermostability profiles. N-glycan cleavage specificity was tested on immunoglobulins bearing different N-glycans, using a mass spectrometry-based approach. Binding affinities of catalytically inactive mutants toward different glycoforms were tested by surface plasmon resonance, and the hydrolytic activities of range of variants were measured to explore determinants of substrate binding. Crystal structures of one of the *B. thetaiotaomicron* paralogues, BT1285, and variants were solved, including one catalytically inactive variant in complex with high-mannose N-glycan (Man₉GlcNAc₂). A very low-resolution (19 Angstrom) TEM complex with a Fc fragment and pendant N-glycan were also presented, supported by other biophysical measurements (SEC, SAXS, etc.); the 1:2 stoichiometry was as expected based on two N-glycans on the Fc protein.

The manuscript is overall well-presented, and contains a copious amount of data and text. Despite its voluminous nature and technical quality, the study does not bring sufficient novelty and insight to merit publication in *Nature Communications*. On one hand, there is already a significant amount of structural data on GH18 N-glycanases from GH18, including oligosaccharide complexes, from human gut bacteria, other bacteria, and other sources (http://www.cazy.org/GH18_structure.html). Also in light of the authors' previous work (refs. 28

and 29), the present study is incremental. On the other hand, the study really does not provide much insight into the biological function of the paralogous enzymes, i.e. it does not clarify why *B. thetaiotaomicron* and other bacteria maintain two independent PULs with overlapping biochemical activities (apart from demonstrating some protein-specific substrate selectivity). As detailed in Comment 5 below, plant N-glycans have not been considered as potential substrates. Finally, as referenced in the text, these two PULs have already previously been the subject of transcriptomic and genetic analyses (ref. 49. It is noted that some additional analysis of existing transcriptomics dataset is presented here). All things considered, the manuscript really amounts to a sound report of detailed biochemical and protein structural data on these two enzymes, and would therefore be suited for publication in a biological chemistry, molecular biology, or structural biology journal.

Some additional comments to improve the manuscript further include:

1. Abstract and throughout: Replace "Bacteroidetes" with the new phylum name "Bacteroidales" and indicate the synonym (syn. Bacteroidetes).

2. lines 74-76: The argument that very few PULs have been studied in detail and that there are a limited number of GH complexes is perhaps an overstatement. Certainly, the limited number of primary references cited (refs. 9 and 10) give a biased impression. See recent reviews and primary literature cited therein: Ndeh D, Gilbert HJ. *FEMS Microbiol Rev.* 2018 Mar 1;42(2):146-164. doi: 10.1093/femsre/fuy002. PMID: 29325042; McKee LS, et al. *Environ Microbiol Rep.* 2021 Oct;13(5):559-581. doi: 10.1111/1758-2229.12980. Epub 2021 Jun 13. PMID: 34036727; Tamura K, Brumer H. *Curr Opin Struct Biol.* 2021 Jun;68:26-40. doi: 10.1016/j.sbi.2020.11.001. Epub 2020 Dec 4. PMID: 33285501

3. line 80: How many is "several"?

4. lines 89-99: The Introduction does not do a particularly good job of describing the types of N-glycans that would normally be found in the human diet. This detracts from the presentation of the work, as it does not appropriately establish context and relevance to the HGM. Lines 553-564 should be moved up and incorporated into the Introduction, possibly replacing or preceding the text on lines 89-99.

5. Introduction and throughout: Plant N-glycans are overlooked as potential substrates for these enzymes, both in the introduction, and experimentally. Analogous to lines 88-99 and 553-564, plant N-glycan structures should be discussed, and activities of these enzymes on representative substrates tested. See Crouch LI, Urbanowicz PA, Baslé A, Cai ZP, Liu L, Voglmeir J, Melo Diaz JM, Benedict ST, Spencer DIR, Bolam DN. *Proc Natl Acad Sci U S A.* 2022 Sep 27;119(39):e2208168119. doi: 10.1073/pnas.2208168119. Epub 2022 Sep 19. PMID: 36122227

6. Lines 186-200: There would be a significant value-addition to the study in demonstrating the non-catalytic, substrate-binding of BT1284, and solving its structure, which the authors are well-equipped to do. Compare with lines 31-334.

7. lines 209-210: Add PDB ID and primary reference.

8. lines 277-279: Explain how these pH stabilities are relevant to the human gut environment.

9. lines 501-502: Are these sequence identity values and SSN cluster based on the whole proteins, or only the GH18 domains? Including non-catalytic domains in the analysis will skew the results.

10. lines 526-527 and Methods: Add GEO accession information (the URL to the GEO database is not particularly helpful).

11. lines 565-572: This text is not logically well-connected to the following text in this paragraph.

12. line 718: "either"? Only one MS is indicated.

End of comments

Reviewer #3:

Remarks to the Author:

This manuscript by D.E. Sastre et al. discusses the discovery and characterization of BT1285, an ENGase that processes high mannose glycans in the human gastrointestinal system. This enzyme was found to catabolize the same N-high mannose glycans as a previously discovered ENGase, BT3987, although under different sets of conditions (namely pH, temperature).

The authors, as part of an impressive international collaborative effort, used many biochemical and bioanalytical techniques and approaches to compare the activity and mechanisms of action of BT1285 vs. BT3987. The authors also identified bacteria other than *Bacteroides thetaiotamicron* that encode similar ENGases.

Overall this article is very clear and well written, and represents an exhaustive amount of well planned and organized work. The manuscript is suitable for publication after a few minor modifications as listed below.

It is too briefly pointed out at lines 455-456 that *A. Finegoldii* bacteria were associated with dysbiosis and other diseases of the intestinal system in humans. Given the scope of this article and of this journal, it would be important to expand this discussion to possible health benefits of the discovery of BT1285 and its mechanisms. Especially that significant similar PUL domains were identified in *A. finegoldii* and *B. Thetaiotamicron*, the authors should discuss the implications of their findings to include medical issues related to intestinal diseases.

2) The acronym "Fc" is used to designate both the flow cell used in SPR and the Fc domain of IgG. The authors should adopt another acronym to describe the flow cell, as note this can lead to confusion.

3) A few acronyms are not defined:

ORFs (line 139); CM5 (line 325); RMSD (line 503); IPTG (line 634); NiNTA (line 639).

4) Reword the sentence of lines 741-742 as follows: "A temperature ramp rate of 1oC/min was used to measure over a temperature rate of 15-95oC."

5) Typos to correct:

Of of (line 238); deglycosylation (line 346); The bottom panel (line 368); residues (line 434); measurements (line 574); filtered (line 638); overnight (line 644); buffers (line 735); Data were (line 812).

Point-by-point response to the reviewers' comments
Manuscript # NCOMS-23-60916

Below please find a point-by-point response to all reviewers' comments concerning the initial submission of our manuscript. Reviewer comments are presented in plain text; **our responses are provided in bold text.**

Reviewer #1 (Remarks to the Author):

This paper by Sastre et al. presented biochemical, biophysical, and structural characterizations of BT1285. BT1285 is shown with ample evidence to be a second HM-processing ENGase (endo- β -N-acetylglucosaminidase) from the GH18 family in *B. thetaiotaomicron*, acting on high-mannose (HM) N-glycans in glycoproteins. They also compared the molecular mechanisms of the two ENGases (BT1285 and BT3987) and identified HM-processing ENGases in other gut bacteria. The functional characterizations to me seem very comprehensive and supporting the conclusions. I have some comments on the methods and presentations

1. Fig 1: authors may add a diagram to show the different Pfam domain architectures of the 4 highlighted SSN clusters (e.g., DUF1735 in BT3987/C2 and DUF4849 in C1/C4). Maybe select one representative protein from each cluster and draw their domain start and end positions and the total protein lengths.

We thank the reviewer for this suggestion. We modified figure 1 according to the reviewer's suggestion. The diagram showing the different domain architectures was added in Figure 1A.

2. Fig 1a: I would like to see EC numbers labeled beside SSN clusters if available. Are there other clusters with ENGase (EC 3.2.1.96) activities beside C2 and C3? Do C2 and C4 have known activities?

We modified Figure 1 according to the reviewer's suggestion. Proteins from the 4 selected clusters in the SSN contain proteins with EC 3.2.1.96. We indicated this in the Figure 1A, but also included a Supplementary Table 6 indicating each EC number present in each network. *B. thetaiotaomicron* also has three GH18 domain-containing proteins with EC 3.2.1.14 with putative chitinase-activity (BT2825, BT1632, BT3050) but these are found in separate SSN clusters.

Besides C1-C4 clusters, there are at least 7 different clusters containing putative ENGases specific for CT N-glycans (as indicated in Supplementary Table 6).

Cluster C2 contains ENGases specific for HM N-glycans (such as EndoF1, EndoH and BT3987). C4 shows higher similarity with C1 (it was split into a new cluster at AST of 40) which has CT N-glycan specificity.

3. CAZy GH18 page shows 53,101 proteins, while line 135 mentioned 36,188 were used for SSN analysis. This might be because the data were downloaded a few years ago. Please supply the time

when you downloaded GH18 data from CAZy. Could you provides reasons why alignment score threshold 50 was used?

We thank the reviewer for this helpful observation. We updated the SSN incorporating all the proteins currently found in CAZy database (February of 2024) which was incorporated in Figure 1A.

Our criteria selection for an alignment score threshold of 50 was in order to reach an alignment stringency that allowed separation into distinct clusters in which proteins shared 35-40% pairwise identity. If the alignment score threshold is too high the network may be overly fractured and isofunctional families will be split into multiple subfamilies. If it is too low, multiple subfamilies will be merged into a single cluster. It has been described in the literature that isofunctional families often share at least 40% seq ID (<https://doi.org/10.1016/j.jmb.2003.08.057>).

Thus, our SSN focuses on significant sequence similarities, reducing noise to increase the reliability of the network, potentially leading to a more focused network of biologically relevant relationships. We performed a sensitivity analysis by testing different threshold values and assessing their impact on the resulting network topology. We used a range of AST from 30 to 65 (see new Figure S1a). The delineation of HM-specific ENGases containing subclusters from our SSN at AST 50, was almost identical to the monophyletic groups inferred from a phylogenetic tree (please see Figure below). Moreover, prior studies (e.g., <https://doi.org/10.1074/jbc.RA119.010619>) employed a threshold of 55 for a similar dataset (GH16 family), obtaining similar number of subfamily clusters.

N-J method Phylogenetic tree of GH18 domains of proteins contained in clusters 2 (BT3987) and 3 (BT1285) composed of GH18 domain HM specific ENGases

The delineation of subfamilies from the SSN was identical to the monophyletic groups inferred from the phylogenetic tree.

4. Paragraph in lines 445-463: some method details are needed to understand how the bioinformatics search in Bacteroidetes phylum was done. It seems to me that the search was limited to PULs in PULDB. Did you use BT1285 and BT3987 proteins to blast against PULs? Would it be possible that some better homologs exist in *Alistipes finegoldii* but not in PULs? For that, you could expand the search to the entire genomes of *Alistipes finegoldii* instead of only its PULs.

The search described in lines 445-463 was limited to PULs because we were focused on trying to find PULs containing ENGases similar to BT1285 and BT3987. We used a tool presented in PULDB to search and identify PULs that are similar to BT1285 and BT3987 PULs. This tool is a PUL modular aligner which allows retrieval of conserved modular organizations. This tool produces local alignments of a query PUL against all PULs in PULDB. It treats each protein relevant to PUL function as one character. (10.1093/nar/gkx1022)

However, we also performed regular BLAST analysis using individual BT1285 and BT397 protein sequences. We found that there are some bacteria that contain orthologues to only one of these ENGases, either BT1285 or BT3987 but not both. We showed the main bacterial HM-specific ENGases in Fig. S6. We mentioned the case of *EfEndo18A* (Q830C5) which

shows 31% of seq. ID with BT1285, whereas it has only 22% ID to BT3987. *EfEndo18A* from *Enterococcus faecalis* was placed in the EndoH-containing cluster (sharing 53% ID with EndoH). Another case is the ENGase from *Mediterranea* sp. An20 (it has 64% ID with BT1285 and only 30% ID with BT3987) (MDR0938643.1).

Moreover, the *A. finegoldii* genome contains 3 GH18 exhibiting similarity to BT1285 (one of those is Alfi_0895, similar to BT1284, an ENGase-like protein), which lacks the typical catalytic residues of ENGases. The other two GH18s similar to BT1285 are Alfi_0894 and Alfi_0882, which are the enzymes that we described in Figure 7. We did not identify any other putative HM-specific ENGase outside those PULs in *A. finegoldii*.

5. Line 530 and Figure S16: provide GEO dataset accessions that are used to make the expression plots.

We provided GEO dataset accessions in Fig. S16 and also included it in the Bioinformatic tools section in Methods.

<https://www.ncbi.nlm.nih.gov/geo/tools/profileGraph.cgi?ID=GDS1849:BT1285> at

<https://www.ncbi.nlm.nih.gov/geo/tools/profileGraph.cgi?ID=GDS1849:BT3987> at

6. Line 587, Figure S13 -> S14?

7. Line 590, PULS -> PULs.

8. Line 615, check the sentence punctuation.

We modified typos in points 6-8 accordingly.

Reviewer #2 (Remarks to the Author):

This manuscript by Sastre et al. presents the biochemical and structural characterisation of Glycoside Hydrolase Family 18 (GH18) N-glycan endo-glycosidases from human gut bacteria, primarily focussing on two paralogues from *Bacteroides thetaiotaomicron*. Biochemical characterisation of a pair of orthologues from *Alistipes finegoldii* is also included. Briefly, a range of classical biochemical data was presented, including pH-activity and thermostability profiles. N-glycan cleavage specificity was tested on immunoglobulins bearing different N-glycans, using a mass spectrometry-based approach. Binding affinities of catalytically inactive mutants toward different glycoforms were tested by surface plasmon resonance, and the hydrolytic activities of range of variants were measured to explore determinants of substrate binding. Crystal structures of one of the *B. thetaiotaomicron* paralogues, BT1285, and variants were solved, including one catalytically inactive variant in complex with high-mannose N-glycan (Man9GlcNAc2). A very low-resolution (19 Angstrom) TEM complex with a Fc fragment and pendant N-glycan were also presented, supported by other biophysical measurements (SEC, SAXS, etc.); the 1:2 stoichiometry was as expected based on two N-glycans on the Fc protein.

The manuscript is overall well-presented and contains a copious amount of data and text. Despite its voluminous nature and technical quality, the study does not bring sufficient novelty and insight to merit publication in Nature Communications.

On one hand, there is already a significant amount of structural data on GH18 N-glycanases from GH18, including oligosaccharide complexes, from human gut bacteria, other bacteria, and other sources (http://www.cazy.org/GH18_structure.html). Also in light of the authors' previous work (refs. 28 and 29), the present study is incremental. On the other hand, the study really does not provide much insight into the biological function of the paralogous enzymes, i.e. it does not clarify why *B. thetaiotaomicron* and other bacteria maintain two independent PULs with overlapping biochemical activities (apart from demonstrating some protein-specific substrate selectivity). As detailed in Comment 5 below, plant N-glycans have not been considered as potential substrates. Finally, as referenced in the text, these two PULs have already previously been the subject of transcriptomic and genetic analyses (ref. 49. It is noted that some additional analysis of existing transcriptomics dataset is presented here). All things considered, the manuscript really amounts to a sound report of detailed biochemical and protein structural data on these two enzymes and would therefore be suited for publication in a biological chemistry, molecular biology, or structural biology journal.

We appreciate the positive comments about our work. As the major concern of our manuscript by Reviewer #2 was an opinion that our work lacked sufficient novelty and insight, we wish to specifically address this overarching concern first, prior to addressing remaining concerns in a point-by-point fashion. We respectfully disagree with this opinion, as we report at least five novel findings that provide unprecedented insight not only into glycan metabolism in the human gut microbiome but to broad biomedical and biotechnological applications, as described below:

- 1. Our study provides the first description of a novel High-Mannose (HM) polysaccharide utilization locus (PUL) in Bacteroidetes that complements the previously described HM PUL (doi.org/10.1038/nature13995). This is the first time that two PULs and their associated carbohydrate-active enzymes targeting the same glycoprotein substrate have been defined mechanistically in one bacterium. Indeed, the data reported in our manuscript are the result of our efforts to determine why some human gut microbes would expend the evolutionary and metabolic energy to maintain two PULs that targeted the same glycan substrate. Were these two HM PULs simply redundant or were there critical mechanistic differences between their gene products, more specifically their GH18 endoglycosidases, that would explain the maintenance of both HM PULs. Indeed, our study demonstrates that these PULs and their associated GH18 endoglycosidases are not at all redundant, due to the following: (i) the products of genes encoding the two GH18 enzymes in these PULs are expressed in different conditions; and (ii) the two GH18 enzymes exhibit different thermal stabilities, optimal pH levels for activity, and glycan binding affinities. Having discovered, described and defined this phenomenon in one bacterium, we wondered if this was unique to *B. thetaiotaomicron* or was, instead, a more general glycan metabolism strategy employed by numerous gut microbes. Indeed, we found that multiple human gut-resident bacteria harbor analogous multiplicative PULs**

encoding HM *N*-glycan-processing endoglycosidases. Thus, our discovery reveals a novel evolutionary strategy employed by these bacteria to efficiently metabolize HM *N*-glycans, an especially relevant finding considering that some of these bacteria are already being used as probiotics or postbiotics, further highlighting the biomedical relevance of our study.

2. Our study also presents the discovery of an entirely new class of proteins or pseudoenzymes: non-catalytic ENGase-like proteins. Following the suggestion of this reviewer, we obtained the first crystal structure of such an ENGase-like protein found in an HM-specific PUL, which we report and describe in this revised manuscript. This protein contains an N-terminal β -sandwich domain (DUF1735) linked to a GH18 domain in which non-conservative mutations in both the active site and putative substrate binding site render it catalytically inactive. Although its function remains unknown, we provide the first description of one member of this new class of proteins with novel insights into its structure and glycan binding properties.
3. We provide the first description of endoglycosidases (and carbohydrate-active enzymes in general) showing different catalytic efficiencies based on glycan accessibility on native glycoprotein substrates. Through our structural and mechanistic studies, we found that the two GH18 endoglycosidases encoded in the two distinct *B. thetaiotaomicron* HM-specific PULs had different abilities to access and hydrolyze HM glycans from distinct quaternary glycoprotein structures, which results in their different catalytic activities. We also elucidated the catalytic mechanism of HM-specific endoglycosidases by identifying specific residues directly involved in the recognition of different glycans.
4. By NS-TEM reconstruction and SAXS we present the first structural model of a protein-agnostic endoglycosidase in complex with its glycoprotein substrate. Although this is a very low-resolution (~ 20 Å) structure, it shows that the mechanism of deglycosylation by protein-agnostic endoglycosidases, in this case BT1285, is notably different than that of protein-specific endoglycosidases (e.g., the IgG-specific endoglycosidase EndoS) for the same glycoprotein substrate. From this structure, we observe that the stoichiometry is 2:1 for the BT1285:IgG Fc region complex, whereas it is 1:1 for the EndoS:IgG Fc region complex. Using intact glycoprotein LC-MS, we found that the accumulation and diminution of the mono-glycosylated IgG Fc species is symmetric over time when this substrate is hydrolyzed by BT1285, as it is not sensing conformational changes. This is in stark contrast to IgG Fc hydrolysis by EndoS. Notably, we were able to obtain the structure of the complex only due to the high affinity between BT1285 and HM glycans, since BT1285 does not bind to protein residues on the IgG Fc region, as is the case for IgG-specific EndoS. Moreover, both IgG-specific peptidases (e.g., IdeS) and Fc γ receptors in complex with IgG Fc regions also form complexes with 1:1 stoichiometry, again different than that observed between for our BT1285:IgG Fc region complex.
5. In this study, we generated an artificial lectin, BT1285i, with high specificity and affinity for HM glycans. It is the first description of an endoglycosidase converted to a glycan-binding protein exhibiting nanomolar affinity to glycans, which could prove highly useful for new and/or improved biomedical and biotechnological applications. Indeed, inactive mutants of endoglycosidases generated in this work (e.g., BT1285i) that recognize different HM *N*-glycans with such high affinity could potentially be

used for targeting tumor-associated HM N-glycans on cell surface receptors in novel immunotherapy applications. Additionally, such artificial lectins could be used to purify therapeutic glycoproteins, such as monoclonal antibodies, which often have some contaminating fraction of HM N-linked glycoforms. Accordingly, based on these discoveries, we filed U.S. Provisional Patent Application No. 63/538,956, on September 18, 2023, entitled “Recombinant Artificial Lectins with Oligomannose Glycan Specificity.” Additionally, these reagents could be applied to modify and/or purify therapeutic glycoproteins beyond antibodies, for diverse biomedical applications.

Regarding the previously reported transcriptomic analysis of these PULs – “Finally, as referenced in the text, these two PULs have already previously been the subject of transcriptomic and genetic analyses (ref. 49. It is noted that some additional analysis of existing transcriptomics dataset is presented here)” – we would like to mention that this was part of a whole *B. thetaiotaomicron* transcriptomic analysis, not a specific and detailed analysis of each of those PULs. We simply used the transcriptomic data to verify the expression of the BT1285 and BT3987 PULs in different conditions *in vivo* and, therefore, we do not believe that it reduces the relevance of our work. If it does, as Reviewer #2 is perhaps suggesting, then by this logic no research on any *B. thetaiotaomicron* gene or gene product could be considered novel, which I am sure we can all agree is not the case.

Some additional comments to improve the manuscript further include:

1. Abstract and throughout: Replace "Bacteroidetes" with the new phylum name "Bacteroidales" and indicate the synonym (syn. Bacteroidetes).

We replaced Bacteroidetes by the new phylum name Bacteroidales (syn. Bacteroidetes) accordingly along the manuscript.

2. lines 74-76: The argument that very few PULs have been studied in detail and that there are a limited number of GH complexes is perhaps an overstatement. Certainly, the limited number of primary references cited (refs. 9 and 10) give a biased impression. See recent reviews and primary literature cited therein:

Ndeh D, Gilbert HJ. FEMS Microbiol Rev. 2018 Mar 1;42(2):146-164. doi: 10.1093/femsre/fuy002. PMID: 29325042; McKee LS, et al. Environ Microbiol Rep. 2021 Oct;13(5):559-581. doi: 10.1111/1758-2229.12980. Epub 2021 Jun 13. PMID: 34036727; Tamura K, Brumer H. Curr Opin Struct Biol. 2021 Jun;68:26-40. doi: 10.1016/j.sbi.2020.11.001. Epub 2020 Dec 4. PMID: 33285501

We modified the paragraph according to the reviewer’s suggestion and we incorporated the recommended citations.

3. line 80: How many is "several"?

The exact number is six (BT1038, BT1044, BT1048, BT1285, BT3987 and BT4709). We added this number to this line of the revised manuscript.

4. lines 89-99: The Introduction does not do a particularly good job of describing the types of N-glycans that would normally be found in the human diet. This detracts from the presentation of the work, as it does not appropriately establish context and relevance to the HGM. Lines 553-564 should be moved up and incorporated into the Introduction, possibly replacing, or preceding the text on lines 89-99.

We modified the Introduction and Discussion according to the reviewer's suggestion.

5. Introduction and throughout: Plant N-glycans are overlooked as potential substrates for these enzymes, both in the introduction, and experimentally. Analogous to lines 88-99 and 553-564, plant N-glycan structures should be discussed, and activities of these enzymes on representative substrates tested. See Crouch LI, Urbanowicz PA, Baslé A, Cai ZP, Liu L, Voglmeir J, Melo Diaz JM, Benedict ST, Spencer DIR, Bolam DN. Proc Natl Acad Sci U S A. 2022 Sep 27;119(39):e2208168119. doi: 10.1073/pnas.2208168119. Epub 2022 Sep 19. PMID: 36122227

We modified the Introduction according to the reviewer's suggestion. We mainly focused on HM N-glycans present in animal, plant and viral glycoproteins.

We also performed additional enzymatic activity measurements using the glycoprotein Horseradish peroxidase (HRP) as a plant complex-type N-glycan substrate. We tested the catalytic activity of BT1285, BT3987, BT1284 and 1284-like against complex type glycans from HRP but, as expected, we did not observe any glycan hydrolysis after overnight incubation. We incorporated this result in Figure S2.

HM-specific enzymes are unable to accommodate either N-glycan structures with α 1,6-fucose attached to the first core GlcNAc (commonly present in mammalian complex-type N-glycans) or α 1,3-fucose attached to the core GlcNAc that are typically decorations in plant and insect N-glycans.

6. Lines 186-200: There would be a significant value-addition to the study in demonstrating the non-catalytic, substrate-binding of BT1284, and solving its structure, which the authors are well-equipped to do. Compare with lines 31-334.

As suggested by the reviewer, we solved the X-ray crystal structure of an ortholog of BT1284 from the related *Bacteroides faecium* sp. nov. (doi.org/10.1099/ijsem.0.005666) at 2.67 and 2.9 Å resolution (Figure S3, Table S4). The protein (A0A6H0KVH8_9BACE, named as 1284-like) exhibits 42%ID, 60%sim with BT1284. It was also inactive in our enzymatic activity assays (against CT and HM substrates, and also CT plant N-glycans from HRP) (Figure S2). We performed surface plasmon resonance (SPR) analysis using RNaseB and IgG-HM as immobilized glycoproteins for BT1284 to experimentally measure N-glycan binding to GH18-like proteins. We detected no binding to these HM glycoproteins (Figure S2). The lack of ENGase activity of BT1284 and 1284-like can be explained by the absence of one of two carboxylate groups (Asp and/or Glu), and by differences in the residues and loops that form the substrate-binding cleft in bona-fide HM specific ENGases (e.g., BT1285, BT3987, EndoH and EndoF1).

7. lines 209-210: Add PDB ID and primary reference.

We added PDB IDs as suggested.

8. lines 277-279: Explain how these pH stabilities are relevant to the human gut environment.

We added this information in the manuscript (lines 582-597).

9. lines 501-502: Are these sequence identity values, and SSN cluster based on the whole proteins, or only the GH18 domains? Including non-catalytic domains in the analysis will skew the results.

Yes, these are sequence identity values. SSN analysis was made using the full-length sequence of ENGases, however, this analysis is based on a local alignment that groups more similar protein domains together. We added this information in the manuscript (lines 626-628). For instance, we observed single domain HM-specific ENGases such as EndoF1 in the cluster containing BT3987 (which has 2 domains), because these only share sequence identity in the GH18 domain. We had performed other SSN analyses and we observed clusters containing both, single and multiple domains proteins together, indicating that the extra non-catalytic domains are not affecting the results.

10. lines 526-527 and Methods: Add GEO accession information (the URL to the GEO database is not particularly helpful).

<https://www.ncbi.nlm.nih.gov/geo/tools/profileGraph.cgi?ID=GDS1849:BT1285> at

<https://www.ncbi.nlm.nih.gov/geo/tools/profileGraph.cgi?ID=GDS1849:BT3987> at

11. lines 565-572: This text is not logically well-connected to the following text in this paragraph.

We agree with the reviewer. We modified this paragraph accordingly (lines 575-601)

12. line 718: "either"? Only one MS is indicated.

It is only one MS. The correct sentence is: The LC system is attached to an Agilent 719 6545XT quadrupole-time of flight (Q-TOF) (Agilent, Santa Clara, CA). We modified the text accordingly.

Reviewer #3 (Remarks to the Author):

This manuscript by D.E. Sastre et al. discusses the discovery and characterization of BT1285, an ENGase that processes high mannose glycans in the human gastrointestinal system. This enzyme was found to catabolize the same N-high mannose glycans as a previously discovered ENGase, BT3987, although under different sets of conditions (namely pH, temperature). The authors, as part of an impressive international collaborative effort, used many biochemical

and bioanalytical techniques and approaches to compare the activity and mechanisms of action of BT1285 vs. BT3987. The authors also identified bacteria other than *Bacteroides thetaiotamicron* that encode similar ENGases.

Overall, this article is very clear and well written and represents an exhaustive amount of well-planned and organized work. The manuscript is suitable for publication after a few minor modifications as listed below.

1) It is too briefly pointed out at lines 455-456 that *A. finegoldii* bacteria were associated with dysbiosis and other diseases of the intestinal system in humans. Given the scope of this article and of this journal, it would be important to expand this discussion to possible health benefits of the discovery of BT1285 and its mechanisms. Especially that significant similar PUL domains were identified in *A. finegoldii* and *B. thetaiotamicron*, the authors should discuss the implications of their findings to include medical issues related to intestinal diseases.

We thank the reviewer for this suggestion. We believe that we have discussed the importance of our findings related to how these proteins secreted from gut bacteria can be directly or indirectly involved in intestinal diseases. However, based on this comment we decided to add new paragraphs in the discussion (please see lines 577-612) about the relevance of glycan metabolism of Bacteroidales in human health.

2) The acronym “Fc” is used to designate both the flow cell used in SPR and the Fc domain of IgG. The authors should adopt another acronym to describe the flow cell, as note this can lead to confusion.

We agree with the reviewer’s suggestion. We replaced Fc acronym in SPR figure 4 and S7 by adding the words Flow Cell 1 and Flow Cell 2.

3) A few acronyms are not defined: ORFs (line 139); CM5 (line 325); RMSD (line 503); IPTG (line 634); NiNTA (line 639).

All the acronyms: Open reading frame (ORF), Carboxy-methylated Biacore Sensor Chip CM5 (CM5), root-mean-square deviation (RMSD), Isopropyl β -d-1-thiogalactopyranoside (IPTG) and Nickel-Nitrilotriacetic acid (Ni-NTA) are now defined in the corresponding section.

4) Reword the sentence of lines 741-742 as follows: “A temperature ramp rate was used to measure over a temperature rate of 15-95.”

We agree with the reviewer’s suggestion. We reworded the sentence accordingly.

5) Typos to correct:

Of of (line 238); deglycosylation (line 346); The bottom panel (line 368); residues (line 434); measurements (line 574); filtered (line 638); overnight (line 644); buffers (line 735); Data were (line 812).

We appreciate the identification of those typos. We fixed all the typos accordingly.

Reviewers' Comments:

Reviewer #1:

Remarks to the Author:

All my comments were appropriately addressed. Thank you.

Reviewer #2:

Remarks to the Author:

In this revision, the authors have satisfactorily addressed my previous comments, and the manuscript is acceptable for publication following additional minor revision, as noted below. The inclusion of the biochemical analysis and structure of the *Bacteroides faecium* "1284-like" protein is an important addition, and I commend the authors on their effort to expand their study as suggested. Ditto for the analysis of activity toward the plant N-glycans on horseradish peroxidase.

One point of clarification is warranted regarding the authors' primary response: I did not suggest that the "work lacked sufficient novelty and insight" and certainly not that "no research on any B. thetaiotaomicron gene or gene product could be considered novel." Rather, I wrote that "the study does not bring sufficient novelty and insight to merit publication *in Nature Communications*" (emphasis added). In this regard, I remain rather unmoved by the authors' passionate response, which is based nonetheless on some fairly thin arguments:

- "first description of a novel High-Mannose (HM) polysaccharide utilization locus (PUL) in Bacteroidetes that complements the previously described HM PUL". The claim of "first" HM PUL not really supported here, especially considering the high degree of synteny between the core proteins from the previously characterised PUL 72 and the presently studied PUL16 (Fig 1b). What does it really mean to say that something is the "first to complement" something that came before?

- "these PULs and their associated GH18 endoglycosidases are not at all redundant, due to the following: (i) the products of genes encoding the two GH18 enzymes in these PULs are expressed in different conditions; and (ii) the two GH18 enzymes exhibit different thermal stabilities, optimal pH levels for activity, and glycan binding affinities." These differences are notable, but the ability of this PUL to enable growth in the absence of PUL72 (lines 130-132) does suggest functional overlap/redundancy. Certainly, this appears to be a case of subtle evolution and specialization (i.e. "subfunctionalization") within the GH18 family.

Regardless, I acknowledge that my opinions of merit are subjective, and I am not interested in a protracted dialog on this point. The work represents a sound and valuable contribution of protein functional and structural information, for reasons stated previously.

Minor revisions:

- The authors should carefully check and revise their spelling and grammar, especially in the new text highlighted in yellow.

- Figure 1: Please distinguish in the legend which 3-D structures are experimental and which are modeled.

- Figure 1b: Please indicate the degree of protein sequence identity and similarity between the syntenic genes in panel b.

- Lines 188-201: A discussion of the lack of activity on HRP as a representative plant N-glycan substrate should be included in the main text, recapitulating the authors' response discussing activity-limiting fucosylation.

- Lines 209-210 and throughout: "1284-like" should be replaced with "*Bacteroides faecium* GH18-like", "*B. faecium* GH18-like", etc.

- Line 59: As the Bacteroidetes are now called Bacteroidales, the Firmicutes are now called

Bacillota. Analogous to line 33, this latter synonym should be noted. See <https://doi.org/10.1099/ijsem.0.005056> and <https://doi.org/10.1099/ijsem.0.002593>

End of comments.

Reviewer #3:

Remarks to the Author:

This referee is satisfied with the authors' modifications and responses. The manuscript is now much improved and ready for publication.

Point-by-point response to the reviewers' comments

Manuscript # NCOMS-23-60916A

Below please find a point-by-point response to all reviewers' comments concerning the submission of our revised manuscript. Reviewer comments are presented in plain text; **our responses are provided in bold text.**

Reviewer #1 (Remarks to the Author):

All my comments were appropriately addressed. Thank you.

We thank the reviewer for contributing to the improvement of the manuscript.

Reviewer #2 (Remarks to the Author):

In this revision, the authors have satisfactorily addressed my previous comments, and the manuscript is acceptable for publication following additional minor revision, as noted below. The inclusion of the biochemical analysis and structure of the *Bacteroides faecium* "1284-like" protein is an important addition, and I commend the authors on their effort to expand their study as suggested. Ditto for the analysis of activity toward the plant N-glycans on horseradish peroxidase.

One point of clarification is warranted regarding the authors' primary response: I did not suggest that the "work lacked sufficient novelty and insight" and certainly not that "no research on any B. thetaiotaomicron gene or gene product could be considered novel." Rather, I wrote that "the study does not bring sufficient novelty and insight to merit publication *in Nature Communications*" (emphasis added). In this regard, I remain rather unmoved by the authors' passionate response, which is based nonetheless on some fairly thin arguments:

- "first description of a novel High-Mannose (HM) polysaccharide utilization locus (PUL) in Bacteroidetes that complements the previously described HM PUL". The claim of "first" HM PUL not really supported here, especially considering the high degree of synteny between the core proteins from the previously characterised PUL 72 and the presently studied PUL16 (Fig 1b). What does it really mean to say that something is the "first to complement" something that came before?

- "these PULs and their associated GH18 endoglycosidases are not at all redundant, due to the following: (i) the products of genes encoding the two GH18 enzymes in these PULs are expressed in different conditions; and (ii) the two GH18 enzymes exhibit different thermal stabilities, optimal pH levels for activity, and glycan binding affinities." These differences are notable, but the ability of this PUL to enable growth in the absence of PUL72 (lines 130-132) does suggest functional overlap/redundancy. Certainly, this appears to be a case of subtle evolution and specialization (i.e. "subfunctionalization") within the GH18 family.

Regardless, I acknowledge that my opinions of merit are subjective, and I am not interested in a protracted dialog on this point. The work represents a sound and valuable contribution of protein functional and structural information, for reasons stated previously.

We thank the reviewer for considering that our work represents a sound and valuable contribution of protein functional and structural information, and for all the constructive comments to improve the manuscript.

Minor revisions:

- The authors should carefully check and revise their spelling and grammar, especially in the new text highlighted in yellow.

We carefully checked the spelling and grammar of the manuscript.

- Figure 1: Please distinguish in the legend which 3-D structures are experimental and which are modeled.

- Figure 1b: Please indicate the degree of protein sequence identity and similarity between the syntenic genes in panel b.

We modified Figure 1b and legend of Figure 1d, accordingly for adding the suggested changes.

- Lines 188-201: A discussion of the lack of activity on HRP as a representative plant N-glycan substrate should be included in the main text, recapitulating the authors' response discussing activity-limiting fucosylation.

We included a paragraph of the lack of activity on plant complex type-N-glycan substrate by HM ENGases (Lines 267-270).

- Lines 209-210 and throughout: "1284-like" should be replaced with "Bacteroides faecium GH18-like", "B. faecium GH18-like", etc.

We replaced 1284-like with *Bacteroides faecium* GH18-like as suggested.

- Line 59: As the Bacteroidetes are now called Bacteroidales, the Firmicutes are now called Bacillota. Analogous to line 33, this latter synonym should be noted. See

<https://doi.org/10.1099/ijsem.0.005056> and <https://doi.org/10.1099/ijsem.0.002593>

We made the suggested change in line 56.

End of comments.

Reviewer #3 (Remarks to the Author):

This referee is satisfied with the authors' modifications and responses. The manuscript is now much improved and ready for publication.

We thank the reviewer for contributing to the improvement of the manuscript.